# Automatic Data Augmentation for Generalization in Reinforcement Learning

**Roberta Raileanu**
New York University
`raileanu@cs.nyu.edu`

**Max Goldstein**
New York University
`mag1038@nyu.edu`

**Denis Yarats**
New York University
Facebook AI Research
`denisyarats@cs.nyu.edu`

**Ilya Kostrikov**
New York University
`kostrikov@cs.nyu.edu`

**Rob Fergus**
New York University
`fergus@cs.nyu.edu`

## Abstract

Deep reinforcement learning (RL) agents often fail to generalize beyond their training environments. To alleviate this problem, recent work has proposed the use of data augmentation. However, different tasks tend to benefit from different types of augmentations and selecting the right one typically requires expert knowledge. In this paper, we introduce three approaches for automatically finding an effective augmentation for any RL task. These are combined with two novel regularization terms for the policy and value function, required to make the use of data augmentation theoretically sound for actor-critic algorithms. Our method achieves a new state-of-the-art[1] on the Procgen benchmark and outperforms popular RL algorithms on DeepMind Control tasks with distractors. In addition, our agent learns policies and representations which are more robust to changes in the environment that are irrelevant for solving the task, such as the background. Our code is available at `https://github.com/rraileanu/auto-drac`.

## 1   Introduction

Generalization to new environments remains a major challenge in deep reinforcement learning (RL). Current methods fail to generalize to unseen environments even when trained on similar settings [19, 51, 71, 11, 21, 12, 60]. This indicates that standard RL agents memorize specific trajectories rather than learning transferable skills. Several strategies have been proposed to alleviate this problem, such as the use of regularization [19, 71, 11, 28], data augmentation [11, 44, 69, 38, 41], or representation learning [72, 74]. In this work, we focus on the use of data augmentation in RL. We identify key differences between supervised learning and reinforcement learning which need to be taken into account when using data augmentation in RL.

More specifically, we show that a naive application of data augmentation can lead to both theoretical and practical problems with standard RL algorithms, such as unprincipled objective estimates and poor performance. As a solution, we propose **Data-regularized Actor-Critic** or **DrAC**, a new algorithm that enables the use of data augmentation with actor-critic algorithms in a theoretically sound way. Specifically, we introduce two regularization terms which constrain the agent's policy and value function to be invariant to various state transformations. Empirically, this approach allows the agent to learn useful behaviors (outperforming strong RL baselines) in settings in which a naive use of data augmentation completely fails or converges to a sub-optimal policy. While we use Proximal Policy Optimization (PPO,  Schulman et al. [56]) to describe and validate our approach, the method

---

[1]in June 2020, at the time of making this work publicly available on arXiv. it has since been surpassed.

can be easily integrated with any actor-critic algorithm with a discrete stochastic policy such as A3C [49], SAC [24], or IMPALA [17].

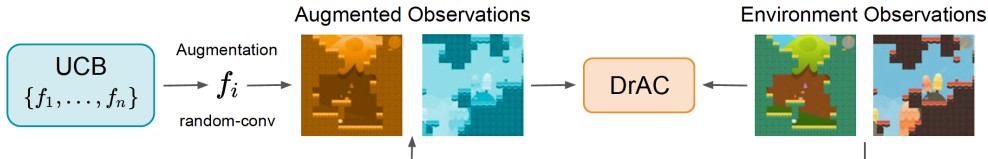

Figure 1: Overview of UCB-DrAC. A UCB bandit selects an image transformation (*e.g.* random-conv) and applies it to the observations. The augmented and original observations are passed to a regularized actor-critic agent (*i.e.* DrAC) which uses them to learn a policy and value function which are invariant to this transformation.

The current use of data augmentation in RL either relies on expert knowledge to pick an appropriate augmentation [11, 44, 38] or separately evaluates a large number of transformations to find the best one [69, 41]. In this paper, we propose three methods for automatically finding a useful augmentation for a given RL task. The first two learn to select the best augmentation from a fixed set, using either a variant of the upper confidence bound algorithm (UCB, Auer [2]) or meta-learning (RL$^2$, Wang et al. [66]). We refer to these methods as **UCB-DrAC** and **RL2-DrAC**, respectively. The third method, **Meta-DrAC**, directly meta-learns the weights of a convolutional network, without access to predefined transformations (MAML, Finn et al. [20]). Figure 1 gives an overview of UCB-DrAC.

We evaluate these approaches on the *Procgen* generalization benchmark [12] which consists of 16 procedurally generated environments with visual observations. Our results show that UCB-DrAC is the most effective among these at finding a good augmentation, and is comparable or better than using DrAC with the best augmentation from a given set. UCB-DrAC also outperforms baselines specifically designed to improve generalization in RL [28, 44, 41] on both train and test. In addition, we show that our agent learns policies and representations that are more invariant to changes in the environment which do not alter the reward or transition function (*i.e.* they are inconsequential for control), such as the background theme.

To summarize, our work makes the following contributions: (i) we introduce a principled way of using data augmentation with actor-critic algorithms, (ii) we propose a practical approach for automatically selecting an effective augmentation in RL settings, (iii) we show that the use of data augmentation leads to policies and representations that better capture task invariances, and (iv) we demonstrate state-of-the-art results on the Procgen benchmark and outperform popular RL methods on four DeepMind Control tasks with natural and synthetic distractors.

## 2 Background

We consider a distribution $q(m)$ of Partially Observable Markov Decision Processes (POMDPs, [6]) $m \in \mathcal{M}$, with $m$ defined by the tuple $(\mathcal{S}_m, \mathcal{O}_m, \mathcal{A}, T_m, R_m, \gamma)$, where $\mathcal{S}_m$ is the state space, $\mathcal{O}_m$ is the observation space, $\mathcal{A}$ is the action space, $T_m(s'|s, a)$ is the transition function, $R_m(s, a)$ is the reward function, and $\gamma$ is the discount factor. During training, we restrict access to a fixed set of POMDPs, $M_{train} = \{m_1, \ldots, m_n\}$, where $m_i \sim q$, $\forall i = \overline{1, n}$. The goal is to find a policy $\pi_\theta$ which maximizes the expected discounted reward over the entire distribution of POMDPs, $J(\pi_\theta) = \mathbb{E}_{q,\pi,T_m,p_m} \left[ \sum_{t=0}^{T} \gamma^t R_m(s_t, a_t) \right]$.

In practice, we use the Procgen benchmark which contains 16 procedurally generated games. Each game corresponds to a distribution of POMDPs $q(m)$, and each level of a game corresponds to a POMDP sampled from that game's distribution $m \sim q$. The POMDP $m$ is determined by the seed (*i.e.* integer) used to generate the corresponding level. Following the setup from Cobbe et al. [12], agents are trained on a fixed set of $n = 200$ levels (generated using seeds from 1 to 200) and tested on the full distribution of levels (generated by sampling seeds uniformly at random from all computer integers).

**Proximal Policy Optimization** (PPO, Schulman et al. [56]) is an actor-critic algorithm that learns a policy $\pi_\theta$ and a value function $V_\theta$ with the goal of finding an optimal policy for a given POMDP.

PPO alternates between sampling data through interaction with the environment and maximizing a clipped surrogate objective function using stochastic gradient ascent.

One component of the PPO objective is the policy gradient term $J_{\mathrm{PG}}$, which is estimated using importance sampling:

$$J_{\mathrm{PG}}(\theta) = \sum_{a \in \mathcal{A}} \pi_\theta(a|s)\hat{A}_{\theta_{\mathrm{old}}}(s,a) = \mathbb{E}_{a \sim \pi_{\theta_{\mathrm{old}}}}\left[\frac{\pi_\theta(a|s)}{\pi_{\theta_{\mathrm{old}}}(a|s)}\hat{A}_{\theta_{\mathrm{old}}}(s,a)\right], \tag{1}$$

where $\hat{A}(\cdot)$ is an estimate of the advantage function, $\pi_{\theta_{\mathrm{old}}}$ is the behavior policy used to collect trajectories (*i.e.* that generates the training distribution of states and actions), and $\pi_\theta$ is the policy we want to optimize (*i.e.* that generates the true distribution of states and actions). See Appendix A for a full description of PPO.

## 3 Automatic Data Augmentation for RL

### 3.1 Data Augmentation in RL

Image augmentation has been successfully applied in computer vision for improving generalization on object classification tasks [58, 9, 8, 39]. As noted by Kostrikov et al. [38], those tasks are invariant to certain image transformations such as rotations or flips, which is not always the case in RL. For example, if your observation is flipped, the corresponding reward will be reversed for the left and right actions and will not provide an accurate signal to the agent. While data augmentation has been previously used in RL settings without other algorithmic changes [11, 69, 41], we argue that this approach is not theoretically sound.

If transformations are naively applied to observations in PPO's buffer, as done in Laskin et al. [41], the PPO objective changes and equation (1) is replaced by

$$J_{\mathrm{PG}}(\theta) = \mathbb{E}_{a \sim \pi_{\theta_{\mathrm{old}}}}\left[\frac{\pi_\theta(a|f(s))}{\pi_{\theta_{\mathrm{old}}}(a|s)}\hat{A}_{\theta_{\mathrm{old}}}(s,a)\right], \tag{2}$$

where $f : \mathcal{S} \times \mathcal{H} \to \mathcal{S}$ is the image transformation. However, the right hand side of the above equation is not a sound estimate of the left hand side because $\pi_\theta(a|f(s)) \neq \pi_\theta(a|s)$, since nothing constrains $\pi_\theta(a|f(s))$ to be close to $\pi_\theta(a|s)$. Note that in the on-policy case when $\theta = \theta_{old}$, the ratio used to estimate the advantage should be equal to one, which is not necessarily the case when using equation (2). In fact, one can define certain transformations $f(\cdot)$ that result in an arbitrarily large ratio $\pi_\theta(a|f(s))/\pi_{\theta_{old}}(a|s)$.

Figure 3 shows examples where a naive use of data augmentation prevents PPO from learning a good policy in practice, suggesting that this is not just a theoretical concern. In the following section, we propose an algorithmic change that enables the use of data augmentation with actor-critic algorithms in a principled way.

### 3.2 Policy and Value Function Regularization

Inspired by the recent work of Kostrikov et al. [38], we propose two novel regularization terms for the policy and value functions that enable the proper use of data augmentation for actor-critic algorithms. Our algorithmic contribution differs from that of Kostrikov et al. [38] in that it constrains both the actor and the critic, as opposed to only regularizing the Q-function. This allows our method to be used with a different (and arguably larger) class of RL algorithms, namely those that learn a policy and a value function.

Following Kostrikov et al. [38], we define an optimality-invariant state transformation $f : \mathcal{S} \times \mathcal{H} \to \mathcal{S}$ as a mapping that preserves both the agent's policy $\pi$ and its value function $V$ such that $V(s) = V(f(s,\nu))$ and $\pi(a|s) = \pi(a|f(s,\nu))$, $\forall s \in \mathcal{S}$, $\nu \in \mathcal{H}$, where $\nu$ are the parameters of $f(\cdot)$, drawn from the set of all possible parameters $\mathcal{H}$.

To ensure that the policy and value functions are invariant to such transformation of the input state, we propose an additional loss term for regularizing the policy,

$$G_\pi = KL\left[\pi_\theta(a|s) \mid \pi_\theta(a|f(s,\nu))\right], \tag{3}$$

as well as an extra loss term for regularizing the value function,

$$G_V = \left(V_\phi(s) - V_\phi\left(f(s, \nu)\right)\right)^2. \tag{4}$$

Thus, our **data-regularized actor-critic** method, or **DrAC**, maximizes the following objective:

$$J_{\text{DrAC}} = J_{\text{PPO}} - \alpha_r(G_\pi + G_V), \tag{5}$$

where $\alpha_r$ is the weight of the regularization term (see Algorithm 1).

The use of $G_\pi$ and $G_V$ ensures that the agent's policy and value function are invariant to the transformations induced by various augmentations. Particular transformations can be used to impose certain inductive biases relevant for the task (*e.g.* invariance with respect to colors or translations). In addition, $G_\pi$ and $G_V$ can be added to the objective of any actor-critic algorithm with a discrete stochastic policy (*e.g.* A3C, TRPO, ACER, SAC, or IMPALA) without any other changes.

Note that when using DrAC, as opposed to the method proposed by Laskin et al. [41], we still use the correct importance sampling estimate of the left hand side objective in equation (1) (instead of a wrong estimate as in equation (2)). This is because the transformed observations $f(s)$ are only used to compute the regularization losses $G_\pi$ and $G_V$, and thus are not used for the main PPO objective. Without these extra terms, the only way to use data augmentation is as explained in Section 3.1, which leads to inaccurate estimates of the PPO objective. Hence, DrAC benefits from the regularizing effect of using data augmentation, while mitigating adverse consequences on the RL objective. See Appendix B for a more detailed explanation of why a naive application of data augmentation with certain policy gradient algorithms is theoretically unsound.

---

**Algorithm 1 DrAC**: **D**ata-**r**egularized **A**ctor-**C**ritic applied to PPO
Black: unmodified actor-critic algorithm.
Cyan: image transformation.
Red: policy regularization.
Blue: value function regularization.

---

1: **Hyperparameters:** image transformation $f$, regularization loss coefficient $\alpha_r$, minibatch size M, replay buffer size T, number of updates K.
2: **for** $k = 1, \ldots, K$ **do**
3:     Collect a new set of transitions $\mathcal{D} = \{(s_i, a_i, r_i, s_{i+1})\}_{i=1}^T$ using $\pi_\theta$.
4:     **for** $j = 1, \ldots, *\frac{T}{M}$ **do**
5:         $\{(s_i, a_i, r_i, s_{i+1})\}_{i=1}^M \sim \mathcal{D}$          ▷ Sample a minibatch of transitions
6:         **for** $i = 1, \ldots, M$ **do**
7:             $\nu_i \sim \mathcal{H}$          ▷ Sample the augmentation parameters
8:             $\hat{\pi}_i \leftarrow \pi_\phi(\cdot|s_i)$          ▷ Compute the policy targets
9:             $\hat{V}_i \leftarrow V_\phi(s_i)$          ▷ Compute the value function targets
10:         **end for**
11:         $G_\pi(\theta) = \frac{1}{M} \sum_{i=1}^M KL\left[\hat{\pi}_i \mid \pi_\theta(\cdot|f(s_i, \nu_i))\right]$          ▷ Regularize the policy
12:         $G_V(\phi) = \frac{1}{M} \sum_{i=1}^M \left(\hat{V}_i - V_\phi\left(f(s_i, \nu_i)\right)\right)^2$          ▷ Regularize the value function
13:         $J_{\text{DrAC}}(\theta, \phi) = J_{\text{PPO}}(\theta, \phi) - \alpha_r(G_\pi(\theta) + G_V(\phi))$    ▷ Compute the total loss function
14:         $\theta \leftarrow \arg\max_\theta J_{\text{DrAC}}$          ▷ Update the policy
15:         $\phi \leftarrow \arg\max_\phi J_{\text{DrAC}}$          ▷ Update the value function
16:     **end for**
17: **end for**

---

## 3.3 Automatic Data Augmentation

Since different tasks benefit from different types of transformations, we would like to design a method that can automatically find an effective transformation for any given task. Such a technique would significantly reduce the computational requirements for applying data augmentation in RL. In this section, we describe three approaches for doing this. In all of them, the augmentation learner is trained at the same time as the agent learns to solve the task using DrAC. Hence, the distribution of rewards varies significantly as the agent improves, making the problem highly nonstationary.

**Upper Confidence Bound.** The problem of selecting a data augmentation from a given set can be formulated as a multi-armed bandit problem, where the action space is the set of available transformations $\mathcal{F} = \{f^1, \ldots, f^n\}$. A popular algorithm for such settings is the upper confidence bound or UCB [2], which selects actions according to the following policy:

$$f_t = \text{argmax}_{f \in \mathcal{F}} \left[ Q_t(f) + c \sqrt{\frac{\log(t)}{N_t(f)}} \right], \tag{6}$$

where $f_t$ is the transformation selected at time step $t$, $N_t(f)$ is the number of times transformation $f$ has been selected before time step $t$ and $c$ is UCB's exploration coefficient. Before the t-th DrAC update, we use equation (6) to select an augmentation $f$. Then, we use equation (5) to update the agent's policy and value function. We also update the counter: $N_t(f) = N_{t-1}(f) + 1$. Next, we collect rollouts with the new policy and update the Q-function: $Q_t(f) = \frac{1}{K} \sum_{i=t-K}^{t} \mathcal{R}(f_i = f)$, which is computed as a sliding window average of the past $K$ mean returns obtained by the agent after being updated using augmentation $f$. We refer to this algorithm as **UCB-DrAC** (Algorithm 4). Note that UCB-DrAC's estimation of $Q(f)$ differs from that of a typical UCB algorithm which uses rewards from the entire history. However, the choice of estimating $Q(f)$ using only more recent rewards is crucial due to the nonstationarity of the problem.

**Meta-Learning the Selection of an Augmentation.** Alternatively, the problem of selecting a data augmentation from a given set can be formulated as a meta-learning problem. Here, we consider a meta-learner like the one proposed by Wang et al. [66]. Before each DrAC update, the meta-learner selects an augmentation, which is then used to update the agent using equation (5). We then collect rollouts using the new policy and update the meta-learner using the mean return of these trajectories. We refer to this approach as **RL2-DrAC** (Algorithm 4).

**Meta-Learning the Weights of an Augmentation.** Another approach for automatically finding an appropriate augmentation is to directly learn the weights of a certain transformation rather than selecting an augmentation from a given set. In this work, we focus on meta-learning the weights of a convolutional network which can be applied to the observations to obtain a perturbed image. We meta-learn the weights of this network using an approach similar to the one proposed by Finn et al. [20]. For each agent update, we also perform a meta-update of the transformation function by splitting DrAC's buffer into meta-train and meta-test sets. We refer to this approach as **Meta-DrAC** (Algorithm 4). More details about the implementation of these methods can be found in Appendix D.

## 4 Experiments

In this section, we evaluate our methods on **four DeepMind Control environments with natural and synthetic distractors** and the **full Procgen benchmark** [12] which consists of 16 procedurally generated games (see Figure 6 in Appendix G). Procgen has a number of attributes that make it a good testbed for generalization in RL: (i) it has a diverse set of games in a similar spirit with the ALE benchmark [5], (ii) each of these games has procedurally generated levels which present agents with meaningful generalization challenges, (iii) agents have to learn motor control directly from images, and (iv) it has a clear protocol for testing generalization.

All environments use a discrete 15 dimensional action space and produce $64 \times 64 \times 3$ RGB observations. We use Procgen's *easy* setup, so for each game, agents are trained on 200 levels and tested on the full distribution of levels. We use PPO as a base for all our methods. More details about our experimental setup and hyperparameters can be found in Appendix E.

**Data Augmentation.** In our experiments, we use a set of eight transformations: *crop, grayscale, cutout, cutout-color, flip, rotate, random convolution* and *color-jitter* [39, 15]. We use **RAD**'s [41] implementation of these transformations, except for *crop*, in which we pad the image with 12 (boundary) pixels on each side and select random crops of $64 \times 64$. We found this implementation of *crop* to be significantly better on Procgen, and thus it can be considered an empirical upper bound of RAD in this case. For simplicity, we will refer to our implementation as RAD. **DrAC** uses the same set of transformations as RAD, but is trained with additional regularization losses for the actor and the critic, as described in Section 3.2.

**Automatic Selection of Data Augmentation.** We compare three different approaches for automatically finding an effective transformation: **UCB-DrAC** which uses UCB [2] to select an augmentation

Table 1: Train and test performance for the Procgen benchmark (aggregated over all 16 tasks, 10 seeds). (a) compares PPO with four baselines specifically designed to improve generalization in RL and shows that they do not significantly help. (b) compares using the best augmentation from our set with and without regularization, corresponding to DrAC and RAD respectively, and shows that regularization improves performance on both train and test. (c) compares different approaches for automatically finding an augmentation for each task, namely using UCB or $RL^2$ for selecting the best transformation from a given set, or meta-learning the weights of a convolutional network (Meta-DrAC). (d) shows additional ablations: DrA regularizes only the actor, DrC regularizes only the critic, Rand-DrAC selects an augmentation using a uniform distribution, Crop-DrAC uses image crops for all tasks, and UCB-RAD is an ablation that does not use the regularization losses. UCB-DrAC performs best on both train and test, and achieves a return comparable with or better than DrAC (which uses the best augmentation).

| | | PPO-Normalized Return (%) | | | | | |
| | | Train | | | Test | | |
| | **Method** | **Median** | **Mean** | **Std** | **Median** | **Mean** | **Std** |
|---|---|---|---|---|---|---|---|
| | PPO | 100.0 | 100.0 | 7.2 | 100.0 | 100.0 | 8.5 |
| (a) | Rand-FM | 93.4 | 87.6 | 8.9 | 91.6 | 78.0 | 9.0 |
| | IBAC-SNI | 91.9 | 103.4 | 8.5 | 86.2 | 102.9 | 8.6 |
| | Mixreg | 95.8 | 104.2 | 3.1 | 105.9 | 114.6 | 3.3 |
| | PLR | 101.5 | 106.7 | 5.6 | 107.1 | 128.3 | 5.8 |
| (b) | **DrAC (Best) (Ours)** | **114.0** | **119.6** | 9.4 | 118.5 | 138.1 | 10.5 |
| | RAD (Best) | 103.7 | 109.1 | 9.6 | 114.2 | 131.3 | 9.4 |
| (c) | **UCB-DrAC (Ours)** | 102.3 | 118.9 | 8.8 | **118.5** | **139.7** | 8.4 |
| | RL2-DrAC | 96.3 | 95.0 | 8.8 | 99.1 | 105.3 | 7.1 |
| | Meta-DrAC | 101.3 | 100.1 | 8.5 | 101.7 | 101.2 | 7.3 |
| (b) | DrA (Best) | 102.6 | 117.7 | 11.1 | 110.8 | 126.6 | 9.0 |
| | DrC (Best) | 103.3 | 108.2 | 10.8 | 110.6 | 115.4 | 8.5 |
| | Rand-DrAC | 100.4 | 99.5 | 8.4 | 102.4 | 103.4 | 7.0 |
| | Crop-DrAC | 97.4 | 112.8 | 9.8 | 114.0 | 132.7 | 11.0 |
| | UCB-RAD | 100.4 | 104.8 | 8.4 | 103.0 | 125.9 | 9.5 |

from a given set, **RL2-DrAC** which uses $RL^2$ [67] to do the same, and **Meta-DrAC** which uses MAML [20] to meta-learn the weights of a convolutional network. Meta-DrAC is implemented using the *higher* library [22]. Note that we do not expect these approaches to be better than DrAC with the best augmentation. In fact, DrAC with the best augmentation can be considered to be an upper bound for these automatic approaches since it uses the best augmentation during the entire training process.

**Ablations. DrC** and **DrA** are ablations to **DrAC** that use only the value or only the policy regularization terms, respectively. DrC can be thought of an analogue of DrQ [38] applied to PPO rather than SAC. **Rand-DrAC** uses a uniform distribution to select an augmentation each time. **Crop-DrAC** uses crop for all games (which is the most effective augmentation on half of the Procgen games). **UCB-RAD** combines UCB with RAD (*i.e.* it does not use the regularization terms).

**Baselines.** We also compare with **Rand-FM** [44], **IBAC-SNI** [28], **Mixreg** [66], and **PLR** [32], four methods specifically designed for improving generalization in RL and previously tested on Procgen environments. Rand-FM uses a random convolutional networks to regularize the learned representations, while IBAC-SNI uses an information bottleneck with selective noise injection. Mixreg uses mixtures of observations to impose linearity constraints between the agent's inputs and the outputs, while PLR samples levels according to their learning potential.

**Evaluation Metrics.** At the end of training, for each method and each game, we compute the average score over 100 episodes and 10 different seeds. The scores are then normalized using the corresponding PPO score on the same game. We aggregate the normalized scores over all 16 Procgen games and report the resulting mean, median, and standard deviation (Table 1). For a per-game breakdown, see Tables 6 and 7 in Appendix J.

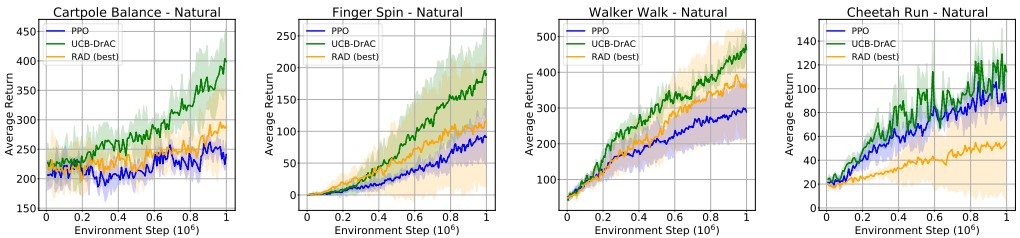

Figure 2: Average return on DMC tasks with natural video backgrounds with mean and standard deviation computed over 5 seeds. UCB-DrAC outperforms PPO and RAD with the best augmentations.

## 4.1 Generalization Performance on Procgen

Table 1 shows train and test performance on Procgen. UCB-DrAC significantly outperforms PPO, Rand-FM, IBAC-SNI, PLR, and Mixreg. As shown in Jiang et al. [32], combining PLR with UCB-DrAC achieves a new state-of-the-art on Procgen leading to a 76% gain over PPO. Regularizing the policy and value function leads to improvements over merely using data augmentation, and thus the performance of DrAC is better than that of RAD (both using the best augmentation for each game). In addition, we demonstrate the importance of regularizing both the policy and the value function rather than either one of them by showing that DrAC is superior to both DrA and DrC (see Figures 9 and 10 in Appendix K). Our experiments show that the most effective way of automatically finding an augmentation is UCB-DrAC. As expected, meta-learning the weights of a CNN using Meta-DrAC performs reasonably well on the games in which the random convolution augmentation helps. But overall, Meta-DrAC and RL2-DrAC are worse than UCB-DrAC. In addition, UCB is generally more stable, easier to implement, and requires less fine-tuning compared to meta-learning algorithms. See Figures 8 and 7 in Appendix K for a comparison of these three approaches on each game. Moreover, automatically selecting the augmentation from a given set using UCB-DrAC performs similarly well or even better than a method that uses the best augmentation for each task throughout the entire training process. UCB-DrAC also achieves higher returns than an ablation that uses a uniform distribution to select an augmentation each time, Rand-DrAC. Nevertheless, UCB-DrAC is better than Crop-DrAC, which uses crop for all the games (which is the best augmentation for eight of the Procgen games as shown in Tables 4 and 5 from Appendix H).

## 4.2 DeepMind Control with Distractors

In this section, we evaluate our approach on the DeepMind Control Suite from pixels (DMC, Tassa et al. [64]). We use four tasks, namely Cartpole Balance, Finger Spin, Walker Walk, and Cheetah Run, in three settings with different types of backgrounds, namely the *default*, *simple* distractors, and *natural* videos from the Kinetics dataset [35], as introduced in Zhang et al. [73]. See Figure 11 for a few examples. Note that in the simple and natural settings, the background is sampled from a list of videos at the beginning of each episode, which creates spurious correlations between the backgrounds and the rewards. In the simple and natural distractor settings, as shown in Figure 13, UCB-DrAC outperforms PPO and RAD on all these environments in the most challenging setting with natural distractors. See Appendix L for results on DMC with default and simple distractor backgrounds, where our method also outperforms the baselines.

## 4.3 Regularization Effect

In Section 3.1, we argued that additional regularization terms are needed in order to make the use of data augmentation in RL theoretically sound. However, one might wonder if this problem actually appears in practice. Thus, we empirically investigate the effect of regularizing the policy and value function. For this purpose, we compare the performance of RAD and DrAC with grayscale and random convolution augmentations on Chaser, Miner, and StarPilot.

Figure 3 shows that not regularizing the policy and value function with respect to the transformations used can lead to drastically worse performance than vanilla RL methods, further emphasizing the

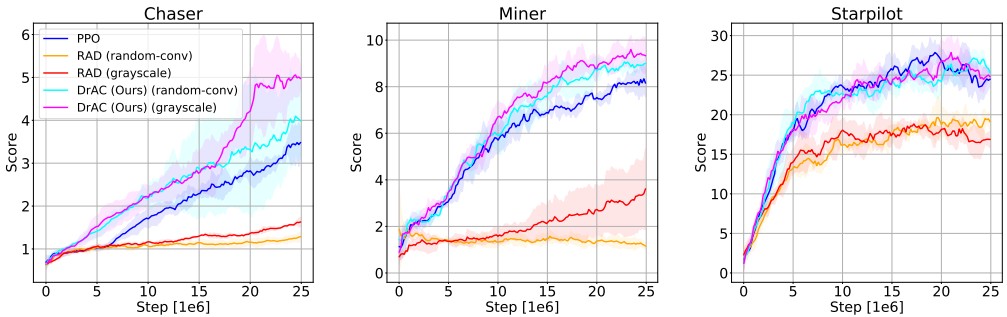

Figure 3: Comparison between RAD and DrAC with the same augmentations, grayscale and random convolution, on the test environments of Chaser (left), Miner (center), and StarPilot (right). While DrAC's performance is comparable or better than PPO's, not using the regularization terms, *i.e.* using RAD, significantly hurts performance relative to PPO. This is because, in contrast to DrAC, RAD does not use a principled (importance sampling) estimate of PPO's objective.

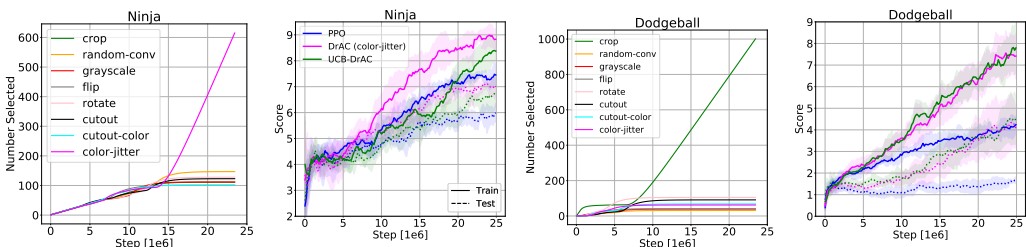

Figure 4: Cumulative number of times UCB selects each augmentation over the course of training for Ninja (a) and Dodgeball (c). Train and test performance for PPO, DrAC with the best augmentation for each game (color-jitter and crop, respectively), and UCB-DrAC for Ninja (b) and Dodgeball (d). UCB-DrAC finds the most effective augmentation from the given set and reaches the performance of DrAC. Our methods improve both train and test performance.

importance of these loss terms. In contrast, using the regularization terms as part of the RL objective (as DrAC does) results in an agent that is comparable or, in some cases, significantly better than PPO.

### 4.4 Automatic Augmentation

Our experiments indicate there is not a single augmentation that works best across all Procgen games (see Tables 4 and 5 in Appendix H). Moreover, our intuitions regarding the best transformation for each game might be misleading. For example, at a first sight, Ninja appears to be somewhat similar to Jumper, but the augmentation that performs best on Ninja is color-jitter, while for Jumper is random-conv (see Tables 4 and 5). In contrast, Miner seems like a different type of game than Climber or Ninja, but they all have the same best performing augmentation, namely color-jitter. These observations further underline the need for a method that can automatically find the right augmentation for each task.

Table 1 along with Figures 8 and 7 in the Appendix compare different approaches for automatically finding an augmentation, showing that UCB-DrAC performs best and reaches the asymptotic performance obtained when the most effective transformation for each game is used throughout the entire training process. Figure 4 illustrates an example of UCB's policy during training on Ninja and Dodgeball, showing that it converges to always selecting the most effective augmentation, namely color-jitter for Ninja and crop for Dodgeball. Finally, Figure 5 in Appendix F illustrates how UCB's behavior varies with its exploration coefficient.

Table 2: JSD and Cycle-Consistency (%) (aggregated across all Procgen tasks) for PPO, RAD and UCB-DrAC, measured between observations that vary only in their background themes (*i.e.* colors and patterns that do not interact with the agent). UCB-DrAC learns more robust policies and representations that are more invariant to changes in the observation that are irrelevant for the task.

| | | | Cycle-Consistency (%) | | | |
| | JSD | | 2-way | | 3-way | |
| Method | Mean | Median | Mean | Median | Mean | Median |
|---|---|---|---|---|---|---|
| PPO | 0.25 | 0.23 | 20.50 | 18.70 | 12.70 | 5.60 |
| RAD | 0.19 | 0.18 | 24.40 | 22.20 | 15.90 | 8.50 |
| UCB-DrAC | **0.16** | **0.15** | **27.70** | **24.80** | **17.30** | **10.30** |

### 4.5 Robustness Analysis

To further investigate the generalizing ability of these agents, we analyze whether the learned policies and state representations are invariant to changes in the observations which are irrelevant for solving the task.

We first measure the Jensen-Shannon divergence (JSD) between the agent's policy for an observation from a training level and a modified version of that observation with a different background theme (*i.e.* color and pattern). Note that the JSD also represents a lower bound for the joint empirical risk across train and test [30]. The background theme is randomly selected from the set of backgrounds available for all other Procgen environments, except for the one of the original training level. Note that the modified observation has the same semantics as the original one (with respect to the reward function), so the agent should have the same policy in both cases. Moreover, many of the backgrounds are not uniform and can contain items such as trees or planets which can be easily misled for objects the agent can interact with. As seen in Table 2, UCB-DrAC has a lower JSD than PPO, indicating that it learns a policy that is more robust to changes in the background.

To quantitatively evaluate the quality of the learned representation, we use the cycle-consistency metric proposed by Aytar et al. [3] and also used by Lee et al. [44]. See Appendix C for more details about this metric. Table 2 reports the percentage of input observations in the seen environment that are cycle-consistent with trajectories in modified unseen environments, which have a different background but the same layout. UCB-DrAC has higher cycle-consistency than PPO, suggesting that it learns representations that better capture relevant task invariances.

## 5   Related Work

**Generalization in Deep RL.** A recent body of work has pointed out the problem of overfitting in deep RL [53, 46, 34, 51, 71, 75, 50, 11, 12, 33, 52, 40, 23]. A promising approach to prevent overfitting is to apply regularization techniques originally developed for supervised learning such as dropout [62, 28] or batch normalization [31, 19, 28]. For example, Igl et al. [28] use selective noise injection with a variational information bottleneck, while Lee et al. [44] regularize the agent's representation with respect to random convolutional transformations. The use of state abstractions has also been proposed for improving generalization in RL [72, 74, 1]. Similarly, Roy and Konidaris [54] and Sonar et al. [59] learn domain-invariant policies via feature alignment, while Stooke et al. [63] decouple representation from policy learning. Igl et al. [29] reduce non-stationarity using policy distillation, while Mazoure et al. [47] maximize the mutual information between the agent's representation of successive time steps. Jiang et al. [32] improve efficiency and generalization by sampling levels according to their learning potential, while Wang et al. [67] use mixtures of observations to impose linearity constraints between the agent's inputs and the outputs. More similar to our work, Cobbe et al. [11], Ye et al. [69] and Laskin et al. [41] add augmented observations to the training buffer of an RL agent. However, as we show here, naively applying data augmentation in RL can lead to both theoretical and practical issues. Our algorithmic contributions alleviate these problems while still benefitting from the regularization effect of data augmentation.

**Data Augmentation** has been extensively used in computer vision for both supervised [42, 4, 43, 58, 9, 10, 39] and self-supervised [16, 48] learning. More recent work uses data augmentation for contrastive learning, leading to state-of-the-art results on downstream tasks [70, 26, 25, 7]. Domain

randomization can also be considered a type of data augmentation, which has proven useful for transferring RL policies from simulation to the real world [65]. However, it requires access to a physics simulator, which is not always available. Recently, a few papers propose the use of data augmentation in RL [11, 44, 61, 38, 41], but all of them use a fixed (set of) augmentation(s) rather than automatically finding the most effective one. The most similar work to ours is that of Kostrikov et al. [38], who propose to regularize the Q-function in Soft Actor-Critic (SAC) [24] using random shifts of the input image. Our work differs from theirs in that it automatically selects an augmentation from a given set, regularizes both the actor and the critic, and focuses on the problem of generalization rather than sample efficiency. While there is a body of work on the automatic use of data augmentation [14, 13, 18, 57, 45], these approaches were designed for supervised learning and, as we explain here, cannot be applied to RL without further algorithmic changes.

## 6 Discussion

In this work, we propose UCB-DrAC, a method for automatically finding an effective data augmentation for RL tasks. Our approach enables the principled use of data augmentation with actor-critic algorithms by regularizing the policy and value functions with respect to state transformations. We show that UCB-DrAC avoids the theoretical and empirical pitfalls typical in naive applications of data augmentation in RL. Our approach improves training performance by $19\%$ and test performance by $40\%$ on the Procgen benchmark, and sets a new state-of-the-art on the Procgen benchmark. In addition, the learned policies and representations are more invariant to spurious correlations between observations and rewards. One limitation of our work is the assumption that using a single type of data augmentation throughout the entire training process is optimal. In future work, we plan to study the effect of using multiple augmentations at different stages during the training of the agent's policy. Another promising avenue for future research is to use a more expressive function class for meta-learning the transformations in order to capture a wider range of inductive biases.

## Acknowledgments and Disclosure of Funding

We would like to thank our NeurIPS reviewers for their valuable feedback on this work. Roberta and Max were supported by the DARPA Machine Commonsense program.

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
