# A PPO

**Proximal Policy Optimization** (PPO) [56] is an actor-critic RL algorithm that learns a policy $\pi_\theta$ and a value function $V_\theta$ with the goal of finding an optimal policy for a given MDP. PPO alternates between sampling data through interaction with the environment and optimizing an objective function using stochastic gradient ascent. At each iteration, PPO maximizes the following objective:

$$J_{\text{PPO}} = J_\pi - \alpha_1 J_V + \alpha_2 S_{\pi_\theta}, \tag{7}$$

where $\alpha_1$, $\alpha_2$ are weights for the different loss terms, $S_{\pi_\theta}$ is the entropy bonus for aiding exploration, $J_V$ is the value function loss defined as

$$J_V = \left(V_\theta(s) - V_t^{\text{target}}\right)^2.$$

The policy objective term $J_\pi$ is based on the policy gradient objective which can be estimated using importance sampling in off-policy settings (*i.e.* when the policy used for collecting data is different from the policy we want to optimize):

$$J_{PG}(\theta) = \sum_{a \in \mathcal{A}} \pi_\theta(a|s)\hat{A}_{\theta_{\text{old}}}(s,a) = \mathbb{E}_{a \sim \pi_{\theta_{\text{old}}}}\left[\frac{\pi_\theta(a|s)}{\pi_{\theta_{\text{old}}}(a|s)}\hat{A}_{\theta_{\text{old}}}(s,a)\right], \tag{8}$$

where $\hat{A}(\cdot)$ is an estimate of the advantage function, $\theta_{old}$ are the policy parameters before the update, $\pi_{\theta_{old}}$ is the behavior policy used to collect trajectories (*i.e.* that generates the training distribution of states and actions), and $\pi_\theta$ is the policy we want to optimize (*i.e.* that generates the true distribution of states and actions).

This objective can also be written as

$$J_{PG}(\theta) = \mathbb{E}_{a \sim \pi_{\theta_{\text{old}}}}\left[r(\theta)\hat{A}_{\theta_{\text{old}}}(s,a)\right], \tag{9}$$

where

$$r_\theta = \frac{\pi_\theta(a|s)}{\pi_{\theta_{\text{old}}}(a|s)}$$

is the importance weight for estimating the advantage function.

PPO is inspired by TRPO [55], which constrains the update so that the policy does not change too much in one step. This significantly improves training stability and leads to better results than vanilla policy gradient algorithms. TRPO achieves this by minimizing the KL divergence between the old (*i.e.* before an update) and the new (*i.e.* after an update) policy. PPO implements the constraint in a simpler way by using a clipped surrogate objective instead of the more complicated TRPO objective. More specifically, PPO imposes the constraint by forcing $r(\theta)$ to stay within a small interval around 1, precisely $[1 - \epsilon, 1 + \epsilon]$, where $\epsilon$ is a hyperparameter. The policy objective term from equation (7) becomes

$$J_\pi = \mathbb{E}_\pi\left[\min\left(r_\theta\hat{A}, \text{ clip}\left(r_\theta, 1 - \epsilon, 1 + \epsilon\right)\hat{A}\right)\right],$$

where $\hat{A} = \hat{A}_{\theta_{\text{old}}}(s,a)$ for brevity. The function clip$(r(\theta), 1 - \epsilon, 1 + \epsilon)$ clips the ratio to be no more than $1 + \epsilon$ and no less than $1 - \epsilon$. The objective function of PPO takes the minimum one between the original value and the clipped version so that agents are discouraged from increasing the policy update to extremes for better rewards.

Note that the use of the Adam optimizer [36] allows loss components of different magnitudes so we can use $G_\pi$ and $G_V$ from equations (3) and (4) to be used as part of the DrAC objective in equation (5) with the same loss coefficient $\alpha_r$. This alleviates the burden of hyperparameter search and means that DrAC only introduces a single extra parameter $\alpha_r$.

# B  Naive Application of Data Augmentation in RL

In this section, we further clarify why a naive application of data augmentation with certain RL algorithms is theoretically unsound. This argument applies for all algorithms that use importance sampling for estimating the policy gradient loss, including TRPO, PPO, IMPALA, or ACER. The use of importance sampling is typically employed when the algorithm performs more than a single policy update using the same data in order to correct for the off-policy nature of the updates. For brevity, we will use PPO to explain this problem.

The correct estimate of the policy gradient objective used in PPO is the one in equation (1) (or equivalently, equation (8)) which does not use the augmented observations at all since we are estimating advantages for the actual observations, $A(s, a)$. The probability distribution used to sample advantages is $\pi_{old}(a|s)$ (rather than $\pi_{old}(a|f(s))$ since we can only interact with the environment via the true observations and not the augmented ones (because the reward and transition functions are not defined for augmented observations). Hence, the correct importance sampling estimate uses $\pi(a|s)/\pi_{old}(a|s)$. Using $\pi(a|f(s))/\pi_{old}(a|f(s))$ instead would be incorrect for the reasons mentioned above. What we argue is that, in the case of RAD, the only way to use the augmented observations $f(s)$ is in the policy gradient objective, whether by $\pi(a|f(s))/\pi_{old}(a|f(s))$ or $\pi(a|f(s))/\pi_{old}(a|s)$, depending on the exact implementation, but both of these are incorrect. In contrast, DrAC does not change the policy gradient objective at all which remains the one in equation (1) and instead uses the augmented observations in the additional regularization losses, as shown in equations (3), (4), and (5).

# C  Cycle-Consistency

Here is a description of the cycle-consistency metric proposed by Aytar et al. [3] and also used in Lee et al. [44] for analyzing the learned representations of RL agents. Given two trajectories $V$ and $U$, $v_i \in V$ first locates its nearest neighbor in the other trajectory $u_j = \text{argmin}_{u \in U} \|h(v_i) - h(u)\|^2$, where $h(\cdot)$ denotes the output of the penultimate layer of trained agents. Then, the nearest neighbor of $u_j \in V$ is located, *i.e.*, $v_k = \text{argmin}_{v \in V} \|h(u_j) - h(u_j)\|_2$, and $v_i$ is defined as cycle-consistent if $|i - k| \leq 1$, *i.e.*, it can return to the original point. Note that this cycle-consistency implies that two trajectories are accurately aligned in the hidden space. Similar to [3], we also evaluate the three-way cycle-consistency by measuring whether vi remains cycle-consistent along both paths, $V \rightarrow U \rightarrow J \rightarrow V$ and $V \rightarrow J \rightarrow U \rightarrow V$, where J is the third trajectory.

# D  Automatic Data Augmentation Algorithms

In this section, we provide more details about the automatic augmentation approaches, as well as pseudocodes for all three methods we propose.

RL2-DrAC uses an LSTM [27] network to select an effective augmentation from a given set, which is used to update the agent's policy and value function according to the DrAC objective from Equation (5). We will refer to this network as a (recurrent) selection policy. The LSTM network takes as inputs the previously selected augmentation and the average return obtained after performing one update of the DrAC agent with this augmentation (using Algorithm 1 with K=1). The LSTM outputs an augmentation from the given set and is rewarded using the average return obtained by the agent after one update with the selected augmentation. The LSTM is trained using REINFORCE [68]. See Algorithm 3 for a pseudocode of RL2-DrAC.

Meta-DrAC meta-learns the weights of a convolutional neural network (CNN) which is used to augment the observations in order to update the agent's policy and value function according to the DrAC objective from Equation (5). For each DrAC update of the agent, we split the trajectories in the replay buffer into meta-train and meta-test using a 9 to 1 ratio. The CNN's weights are updated using MAML [20] where the objective function is maximizing the average return obtained by DrAC (after an update using Algorithm 1 with $K = 1$ and the CNN as the transformation $f$).

---

**Algorithm 2 UCB-DrAC**

---

1: **Hyperparameters:** Set of image transformations $\mathcal{F} = \{f^1, \dots, f^n\}$, exploration coefficient c, window for estimating the Q-functions W, number of updates K, initial policy parameters $\pi_\theta$, initial value function $V_\phi$.
2: $N(f) = 1, \ \forall \ f \in \mathcal{F}$ ▷ Initialize the number of times each augmentation was selected
3: $Q(f) = 0, \ \forall \ f \in \mathcal{F}$ ▷ Initialize the Q-functions for all augmentations
4: $R(f) = \text{FIFO}(W), \ \forall \ f \in \mathcal{F}$ ▷ Initialize the lists of returns for all augmentations
5: **for** $k = 1, \dots, K$ **do**
6:     $f_k = \text{argmax}_{f \in \mathcal{F}} \left[ Q(f) + c \sqrt{\frac{\log(k)}{N(f)}} \right]$ ▷ Use UCB to select an augmentation
7:     Update the policy and value function according to Algorithm 1 with $f = f_k$ and $K = 1$:
8:     $\theta \leftarrow \arg\max_\theta J_{\text{DrAC}}$ ▷ Update the policy
9:     $\phi \leftarrow \arg\max_\phi J_{\text{DrAC}}$ ▷ Update the value function
10:     Compute the mean return obtained by the new policy $r_k$.
11:     Add $r_k$ to the $R(f_k)$ list using the first-in-first-out rule.
12:     $Q(f_k) \leftarrow \frac{1}{|R(f_k)|} \sum_{r \in R(f_k)} r$
13:     $N(f_k) \leftarrow N(f_k) + 1$
14: **end for**

---

---

**Algorithm 3 RL2-DrAC**

---

1: **Hyperparameters:** Set of image transformations $\mathcal{F} = \{f^1, \dots, f^n\}$, number of updates K, initial policy $\pi_\theta$, initial value function $V_\phi$.
2: Initialize the selection poicy as an LSTM network $g$ with parameters $\psi$.
3: $f_0 \sim \mathcal{F}$ ▷ Randomly initialize the augmentation
4: $r_0 \leftarrow 0$ ▷ Initialize the average return
5: **for** $k = 1, \dots, K$ **do**
6:     $f_k \sim g_\psi(f_{k-1}, r_{k-1})$ ▷ Select an augmentation according to the recurrent policy
7:     Update the policy and value function according to Algorithm 1 with $f = f_k$ and $K = 1$:
8:     $\theta \leftarrow \arg\max_\theta J_{\text{DrAC}}$ ▷ Update the policy
9:     $\phi \leftarrow \arg\max_\phi J_{\text{DrAC}}$ ▷ Update the value function
10:     Compute the mean return obtained by the new policy $r_k$.
11:     Reward $g_\psi$ with $r_k$.
12:     Update $g_\psi$ using REINFORCE.
13: **end for**

---

---

**Algorithm 4 Meta-DrAC**

---

1: **Hyperparameters:** Distribution over tasks (or levels) $q(m)$, number of updates K, step size parameters $\alpha$ and $\beta$, initial policy $\pi_\theta$, initial value function $V_\phi$.
2: Initialize the set of all training levels $\mathcal{D} = \{l\}_{i=1}^L$.
3: Initialize the augmentation as a CNN $g$ with parameters $\psi$.
4: **for** $k = 1, \dots, K$ **do**
5:     Sample a batch of tasks $m_i \sim q(m)$
6:     **for all** $m_i$ **do**
7:         Collect trajectories on task $m_i$ using the current policy.
8:         Update the policy and value function according to Algorithm 1 with $f = f_k$ and $K = 1$:
9:         $\theta \leftarrow \arg\max_\theta J_{\text{DrAC}}$ ▷ Update the policy
10:         $\phi \leftarrow \arg\max_\phi J_{\text{DrAC}}$ ▷ Update the value function
11:         Compute the return of the new agent on task $m_i$ after being updated with $g_\psi$, $r_{m_i}(g_\psi)$
12:         $\psi_i' \leftarrow \psi + \alpha \nabla_\psi r_{m_i}(g_\psi)$
13:     **end for**
14:     $\psi \leftarrow \psi + \beta \nabla_\psi \sum_{m_i \sim q(m)} r_{m_i}(g_{\psi_i'})$
15: **end for**

---

Table 3: List of hyperparameters used to obtain the results in this paper.

| Hyperparameter | Value |
|---|---|
| $\gamma$ | 0.999 |
| $\lambda$ | 0.95 |
| # timesteps per rollout | 256 |
| # epochs per rollout | 3 |
| # minibatches per epoch | 8 |
| entropy bonus | 0.01 |
| clip range | 0.2 |
| reward normalization | yes |
| learning rate | 5e-4 |
| # workers | 1 |
| # environments per worker | 64 |
| # total timesteps | 25M |
| optimizer | Adam |
| LSTM | no |
| frame stack | no |
| $\alpha_r$ | 0.1 |
| c | 0.1 |
| K | 10 |

# E   Hyperparameters

We use Kostrikov [37]'s implementation of PPO [56], on top of which all our methods are build. The agent is parameterized by the ResNet architecture from [17] which was used to obtain the best results in Cobbe et al. [12]. Following Cobbe et al. [12], we also share parameters between the policy an value networks. To improve stability when training with DrAC, we only backpropagate gradients through $\pi(a|f(s, \nu))$ and $V(f(s, \nu))$ in equations (3) and (4), respectively. Unless otherwise noted, we use the best hyperparameters found in Cobbe et al. [12] for the easy mode of Procgen (*i.e.* same experimental setup as the one used here), namely:

We use the Adam [36] optimizer for all our experiments. Note that by using Adam, we do not need separate coefficients for the policy and value regularization terms (since Adam rescales gradients for each loss component accordingly).

For DrAC, we did a grid search for the regularization coefficient $\alpha_r \in [0.0001, 0.01, 0.05, 0.1, 0.5, 1.0]$ used in equation (5) and found that the best value is $\alpha_r = 0.1$, which was used to produce all the results in this paper.

For UCB-DrAC, we did grid searches for the exploration coefficient $c \in [0.0, 0.1, 0.5, 1.0, 5.0]$ and the size of the sliding window used to compute the Q-values $K \in [10, 50, 100]$. We found that the best values are $c = 0.1$ and $K = 10$, which were used to obtain the results shown here.

For RL2-DrAC, we performed a hyperparameter search for the dimension of recurrent hidden state $h \in [16, 32, 64]$, for the learning rate $l \in [3e - 4, 5e - 4, 7e - 4]$, and for the entropy coefficient $e \in [1e - 4, 1e - 3, 1e - 2]$ and found $h = 32$, $l = 5e - 4$, and $e = 1e - 3$ to work best. We used Adam with $\epsilon = 1e - 5$ as the optimizer.

For Meta-DrAC, the convolutional network whose weights we meta-learn consists of a single convolutional layer with 3 input and 3 output channels, kernel size 3, stride 1 and 0 padding. At each epoch, we perform one meta-update where we unroll the inner optimizer using the training set and compute the meta-test return on the validation set. We did the same hyperparameter grid searches as for RL2-DrAC and found that the best values were $l = 7e - 4$ and $e = 1e - 2$ in this case. The buffer of experience (collected before each PPO update) was split into 90% for meta-training and 10% for meta-testing.

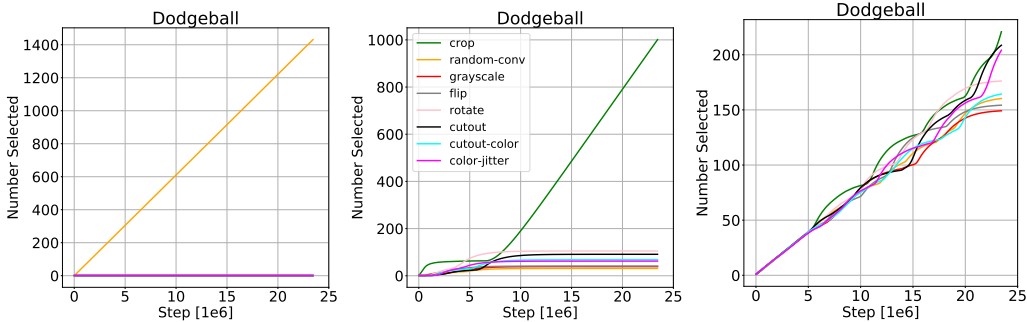

Figure 5: Behavior of UCB for different values of its exploration coefficient c on Dodgeball. When c is too small, UCB might converge to a suboptimal augmentation. On the other hand, when c is too large, UCB might take too long to converge.

For Rand-FM [44] we use the recommended hyperparameters in the authors' released implementation, which were the best values for CoinRun [11], one of the Procgen games used for evaluation in [44].

For IBAC-SNI [28] we also use the authors' open sourced implementation. We use the parameters corresponding to IBAC-SNI $\lambda = .5$. We use weight regularization with $l_2 = .0001$, data augmentation turned on, and a value of $\beta = .0001$ which turns on the variational information bottleneck, and selective noise injection turned on. This corresponds to the best version of this approach, as found by the authors after evaluating it on CoinRun [11]. While IBAC-SNI outperforms the other methods on maze-like games such as heist, maze, and miner, it is still significantly worse than our approach on the entire Procgen benchmark.

For both baselines, Rand-FM and IBAC-SNI, we use the same experimental setup for training and testing as the one used for our methods. Hence, we train them for 25M frames on the easy mode of each Procgen game, using (the same) 200 levels for training and the rest of the levels for testing.

## F  Analysis of UCB's Behavior

In Figure 4, we show the behavior of UCB during training, along with train and test performance on the respective environments. In the case of Ninja, UCB converges to always selecting the best augmentation only after 15M training steps. This is because the augmentations have similar effects on the agent early in training, so it takes longer to find the best augmentation from the given set. In contrast, on Dodgeball, UCB finds the most effective augmentation much earlier in training because there is a significant difference between the effect of various augmentations. Early discovery of an effective augmentation leads to significant improvements over PPO, for both train and test environments.

Another important factor is the exploration coefficient used by UCB (see equation (6)) to balance the exploration and exploitation of different augmentations. Figure 5 compares UCB's behavior for different values of the exploration coefficient. Note that if the coefficient is 0, UCB always selects the augmentation with the largest Q-value. This can sometimes lead to UCB converging on a suboptimal augmentation due to the lack of exploration. However, if the exploration term of equation (6) is too large relative to the differences in the Q-values among various augmentations, UCB might take too long to converge. In our experiments, we found that an exploration coefficient of 0.1 results in a good exploration-exploitation balance and works well across all Procgen games.

## G Procgen Benchmark

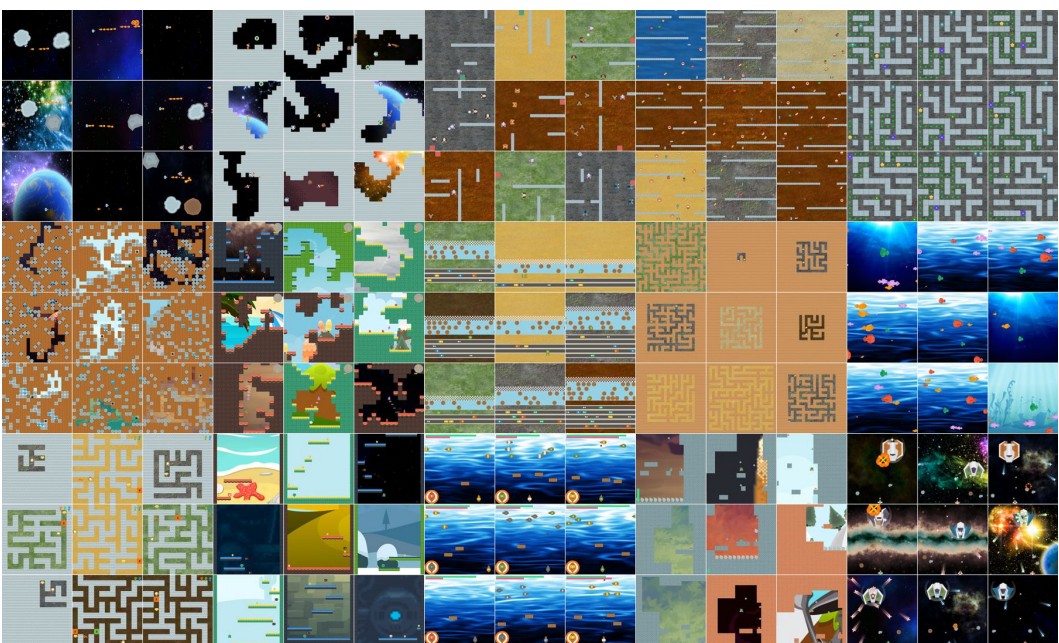

Figure 6: Screenshots of multiple procedurally-generated levels from 15 Procgen environments: StarPilot, CaveFlyer, Dodgeball, FruitBot, Chaser, Miner, Jumper, Leaper, Maze, BigFish, Heist, Climber, Plunder, Ninja, BossFight (from left to right, top to bottom).

## H Best Augmentations

Table 4: Best augmentation type for each game, as evaluated on the test environments.

| Game | BigFish | StarPilot | FruitBot | BossFight | Ninja | Plunder | CaveFlyer | CoinRun |
|---|---|---|---|---|---|---|---|---|
| **Best Augmentation** | crop | crop | crop | flip | color-jitter | crop | rotate | random-conv |

Table 5: Best augmentation type for each game, as evaluated on the test environments.

| Game | Jumper | Chaser | Climber | Dodgeball | Heist | Leaper | Maze | Miner |
|---|---|---|---|---|---|---|---|---|
| **Best Augmentation** | random-conv | crop | color-jitter | crop | crop | crop | crop | color-jitter |

## I Computational Resources

Each of our experiments were run on a single NVIDIA GTX 1080 GPU and 8 CPUs. Trainig UCB-DrAC on one of the Procgen games for 25m steps takes about 7 hours on this hardware. In contrast, one experiment on DMC takes about 1 hour. To thoroughly compare our approach to all relevant methods and ablations, we ran 29 models (see Table 1, with RAD and DrAC run with each of the 8 augmentations) on 16 Procgen games and 3 models on 12 DMC tasks, 10 seeds each, leading to 5000 experiments, each taking 1-8 hours on a GPU. In total, this amounts to approximately 25000 hours of GPU needed to run all the experiments for this paper.

## J  Breakdown of Procgen Scores

Table 6: Procgen scores on train levels after training on 25M environment steps. The mean and standard deviation are computed using 10 runs. The best augmentation for each game is used when computing the results for DrAC and RAD.

| Game | PPO | Rand + FM | IBAC-SNI | DrAC | RAD | UCB-DrAC | RL2-DrAC | Meta-DrAC |
|---|---|---|---|---|---|---|---|---|
| BigFish | $8.9 \pm 1.5$ | $6.0 \pm 0.9$ | $\mathbf{19.1 \pm 0.8}$ | $13.1 \pm 2.2$ | $13.2 \pm 2.8$ | $13.2 \pm 2.2$ | $10.1 \pm 1.9$ | $9.28 \pm 1.9$ |
| StarPilot | $29.8 \pm 2.3$ | $26.3 \pm 0.8$ | $26.7 \pm 0.7$ | $\mathbf{38.0 \pm 3.1}$ | $36.5 \pm 3.9$ | $35.3 \pm 2.2$ | $30.6 \pm 2.6$ | $30.5 \pm 3.9$ |
| FruitBot | $29.1 \pm 1.1$ | $29.2 \pm 0.7$ | $29.4 \pm 0.8$ | $29.4 \pm 1.0$ | $26.1 \pm 3.0$ | $\mathbf{29.5 \pm 1.2}$ | $29.2 \pm 1.0$ | $29.4 \pm 1.1$ |
| BossFight | $\mathbf{8.5 \pm 0.7}$ | $5.6 \pm 0.7$ | $7.9 \pm 0.7$ | $8.2 \pm 1.0$ | $8.1 \pm 1.1$ | $8.2 \pm 0.8$ | $8.4 \pm 0.8$ | $7.9 \pm 0.5$ |
| Ninja | $7.4 \pm 0.7$ | $7.2 \pm 0.6$ | $8.3 \pm 0.8$ | $8.8 \pm 0.5$ | $\mathbf{8.9 \pm 0.9}$ | $8.5 \pm 0.3$ | $8.1 \pm 0.6$ | $7.8 \pm 0.4$ |
| Plunder | $6.0 \pm 0.5$ | $5.5 \pm 0.7$ | $6.0 \pm 0.6$ | $9.9 \pm 1.3$ | $8.4 \pm 1.5$ | $\mathbf{11.1 \pm 1.6}$ | $5.3 \pm 0.5$ | $6.5 \pm 0.5$ |
| CaveFlyer | $6.8 \pm 0.6$ | $6.5 \pm 0.5$ | $6.2 \pm 0.5$ | $\mathbf{8.2 \pm 0.7}$ | $6.0 \pm 0.8$ | $5.7 \pm 0.6$ | $5.3 \pm 0.8$ | $6.5 \pm 0.7$ |
| CoinRun | $9.3 \pm 0.3$ | $9.6 \pm 0.6$ | $9.6 \pm 0.4$ | $\mathbf{9.7 \pm 0.2}$ | $9.6 \pm 0.4$ | $9.5 \pm 0.3$ | $9.1 \pm 0.3$ | $9.4 \pm 0.2$ |
| Jumper | $8.3 \pm 0.4$ | $8.9 \pm 0.4$ | $8.5 \pm 0.6$ | $\mathbf{9.1 \pm 0.4}$ | $8.6 \pm 0.4$ | $8.1 \pm 0.7$ | $8.6 \pm 0.4$ | $8.4 \pm 0.5$ |
| Chaser | $4.9 \pm 0.5$ | $2.8 \pm 0.7$ | $3.1 \pm 0.8$ | $\mathbf{7.1 \pm 0.5}$ | $6.4 \pm 1.0$ | $7.6 \pm 1.0$ | $4.5 \pm 0.7$ | $5.5 \pm 0.8$ |
| Climber | $8.4 \pm 0.8$ | $7.5 \pm 0.8$ | $7.1 \pm 0.7$ | $\mathbf{9.9 \pm 0.8}$ | $9.3 \pm 1.1$ | $9.0 \pm 0.4$ | $7.9 \pm 0.9$ | $8.5 \pm 0.5$ |
| Dodgeball | $4.2 \pm 0.5$ | $4.3 \pm 0.3$ | $\mathbf{9.4 \pm 0.6}$ | $7.5 \pm 1.0$ | $5.0 \pm 0.7$ | $8.3 \pm 0.9$ | $6.3 \pm 1.1$ | $4.8 \pm 0.6$ |
| Heist | $\mathbf{7.1 \pm 0.5}$ | $6.0 \pm 0.5$ | $4.8 \pm 0.7$ | $6.8 \pm 0.7$ | $6.2 \pm 0.9$ | $6.9 \pm 0.4$ | $5.6 \pm 0.8$ | $6.6 \pm 0.6$ |
| Leaper | $\mathbf{5.5 \pm 0.4}$ | $3.2 \pm 0.7$ | $2.7 \pm 0.4$ | $5.0 \pm 0.7$ | $4.9 \pm 0.9$ | $5.3 \pm 0.5$ | $2.7 \pm 0.6$ | $3.7 \pm 0.6$ |
| Maze | $9.1 \pm 0.3$ | $8.9 \pm 0.6$ | $8.2 \pm 0.8$ | $8.3 \pm 0.7$ | $8.4 \pm 0.7$ | $8.7 \pm 0.6$ | $7.0 \pm 0.7$ | $\mathbf{9.2 \pm 0.2}$ |
| Miner | $12.2 \pm 0.3$ | $11.7 \pm 0.8$ | $8.5 \pm 0.7$ | $12.5 \pm 0.3$ | $\mathbf{12.6 \pm 1.0}$ | $12.5 \pm 0.2$ | $10.9 \pm 0.5$ | $12.4 \pm 0.3$ |

Table 7: Procgen scores on test levels after training on 25M environment steps. The mean and standard deviation are computed using 10 runs. The best augmentation for each game is used when computing the results for DrAC and RAD.

| Game | PPO | Rand + FM | IBAC-SNI | DrAC | RAD | UCB-DrAC | RL2-DrAC | Meta-DrAC |
|---|---|---|---|---|---|---|---|---|
| BigFish | $4.0 \pm 1.2$ | $0.6 \pm 0.8$ | $0.8 \pm 0.9$ | $8.7 \pm 1.4$ | $\mathbf{9.9 \pm 1.7}$ | $9.7 \pm 1.0$ | $6.0 \pm 0.5$ | $3.3 \pm 0.6$ |
| StarPilot | $24.7 \pm 3.4$ | $8.8 \pm 0.7$ | $4.9 \pm 0.8$ | $29.5 \pm 5.4$ | $\mathbf{33.4 \pm 5.1}$ | $30.2 \pm 2.8$ | $29.4 \pm 2.0$ | $26.6 \pm 2.8$ |
| FruitBot | $26.7 \pm 0.8$ | $24.5 \pm 0.7$ | $24.7 \pm 0.8$ | $28.2 \pm 0.8$ | $27.3 \pm 1.8$ | $\mathbf{28.3 \pm 0.9}$ | $27.5 \pm 1.6$ | $27.4 \pm 0.8$ |
| BossFight | $7.7 \pm 1.0$ | $1.7 \pm 0.9$ | $1.0 \pm 0.7$ | $7.5 \pm 0.8$ | $7.9 \pm 0.6$ | $\mathbf{8.3 \pm 0.8}$ | $7.6 \pm 0.9$ | $7.7 \pm 0.7$ |
| Ninja | $5.9 \pm 0.7$ | $6.1 \pm 0.4$ | $\mathbf{9.2 \pm 0.6}$ | $7.0 \pm 0.4$ | $6.9 \pm 0.4$ | $6.9 \pm 0.6$ | $6.2 \pm 0.5$ | $5.9 \pm 0.7$ |
| Plunder | $5.0 \pm 0.5$ | $3.0 \pm 0.6$ | $2.1 \pm 0.8$ | $\mathbf{9.5 \pm 1.0}$ | $8.5 \pm 1.2$ | $8.9 \pm 1.0$ | $4.6 \pm 0.3$ | $5.6 \pm 0.4$ |
| CaveFlyer | $5.1 \pm 0.9$ | $5.4 \pm 0.8$ | $\mathbf{8.0 \pm 0.8}$ | $6.3 \pm 0.8$ | $5.1 \pm 0.6$ | $5.3 \pm 0.9$ | $4.1 \pm 0.9$ | $5.5 \pm 0.4$ |
| CoinRun | $8.5 \pm 0.5$ | $\mathbf{9.3 \pm 0.4}$ | $8.7 \pm 0.6$ | $8.8 \pm 0.$ | $9.0 \pm 0.8$ | $8.5 \pm 0.6$ | $8.3 \pm 0.5$ | $8.6 \pm 0.5$ |
| Jumper | $5.8 \pm 0.5$ | $5.3 \pm 0.6$ | $3.6 \pm 0.6$ | $\mathbf{6.6 \pm 0.4}$ | $6.5 \pm 0.6$ | $6.4 \pm 0.6$ | $6.5 \pm 0.5$ | $5.8 \pm 0.7$ |
| Chaser | $5.0 \pm 0.8$ | $1.4 \pm 0.7$ | $1.3 \pm 0.5$ | $5.7 \pm 0.6$ | $5.9 \pm 1.0$ | $\mathbf{6.7 \pm 0.6}$ | $3.8 \pm 0.5$ | $5.1 \pm 0.6$ |
| Climber | $5.7 \pm 0.8$ | $5.3 \pm 0.7$ | $3.3 \pm 0.6$ | $\mathbf{7.1 \pm 0.7}$ | $6.9 \pm 0.8$ | $6.5 \pm 0.8$ | $6.3 \pm 0.5$ | $6.6 \pm 0.6$ |
| Dodgeball | $\mathbf{11.7 \pm 0.3}$ | $0.5 \pm 0.4$ | $1.4 \pm 0.4$ | $4.3 \pm 0.8$ | $2.8 \pm 0.7$ | $4.7 \pm 0.7$ | $3.0 \pm 0.8$ | $1.9 \pm 0.5$ |
| Heist | $2.4 \pm 0.5$ | $2.4 \pm 0.6$ | $\mathbf{9.8 \pm 0.6}$ | $4.0 \pm 0.8$ | $4.1 \pm 1.0$ | $4.0 \pm 0.7$ | $2.4 \pm 0.4$ | $2.0 \pm 0.6$ |
| Leaper | $4.9 \pm 0.7$ | $6.2 \pm 0.5$ | $\mathbf{6.8 \pm 0.6}$ | $5.3 \pm 1.1$ | $4.3 \pm 1.0$ | $5.0 \pm 0.3$ | $2.8 \pm 0.7$ | $3.3 \pm 0.4$ |
| Maze | $5.7 \pm 0.6$ | $8.0 \pm 0.7$ | $\mathbf{10.0 \pm 0.7}$ | $6.6 \pm 0.8$ | $6.1 \pm 1.0$ | $6.3 \pm 0.6$ | $5.6 \pm 0.3$ | $5.2 \pm 0.6$ |
| Miner | $8.5 \pm 0.5$ | $7.7 \pm 0.6$ | $8.0 \pm 0.6$ | $\mathbf{9.8 \pm 0.6}$ | $9.4 \pm 1.2$ | $9.7 \pm 0.7$ | $8.0 \pm 0.4$ | $9.2 \pm 0.7$ |

# K Procgen Learning Curves

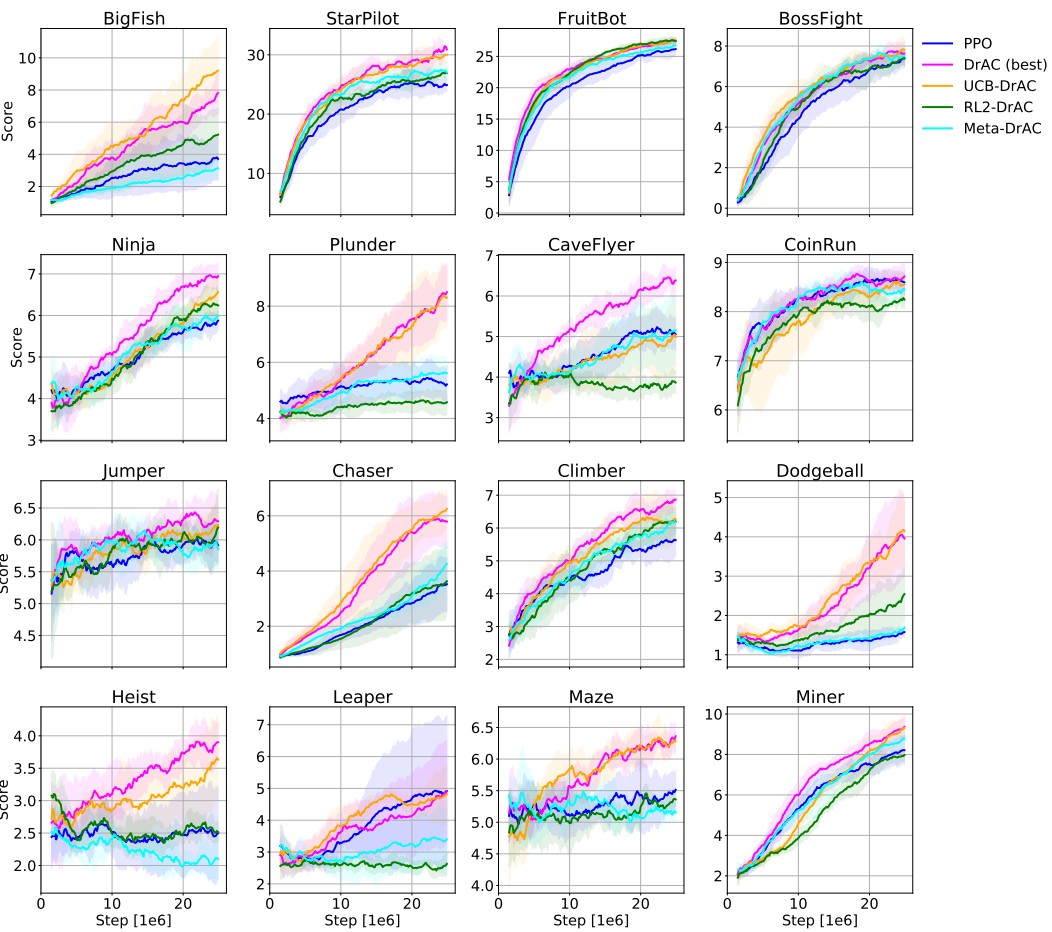

Figure 7: Test performance of various approaches that automatically select an augmentation, namely UCB-DrAC, RL2-DrAC, and Meta-DrAC. The mean and standard deviation are computed using 10 runs.

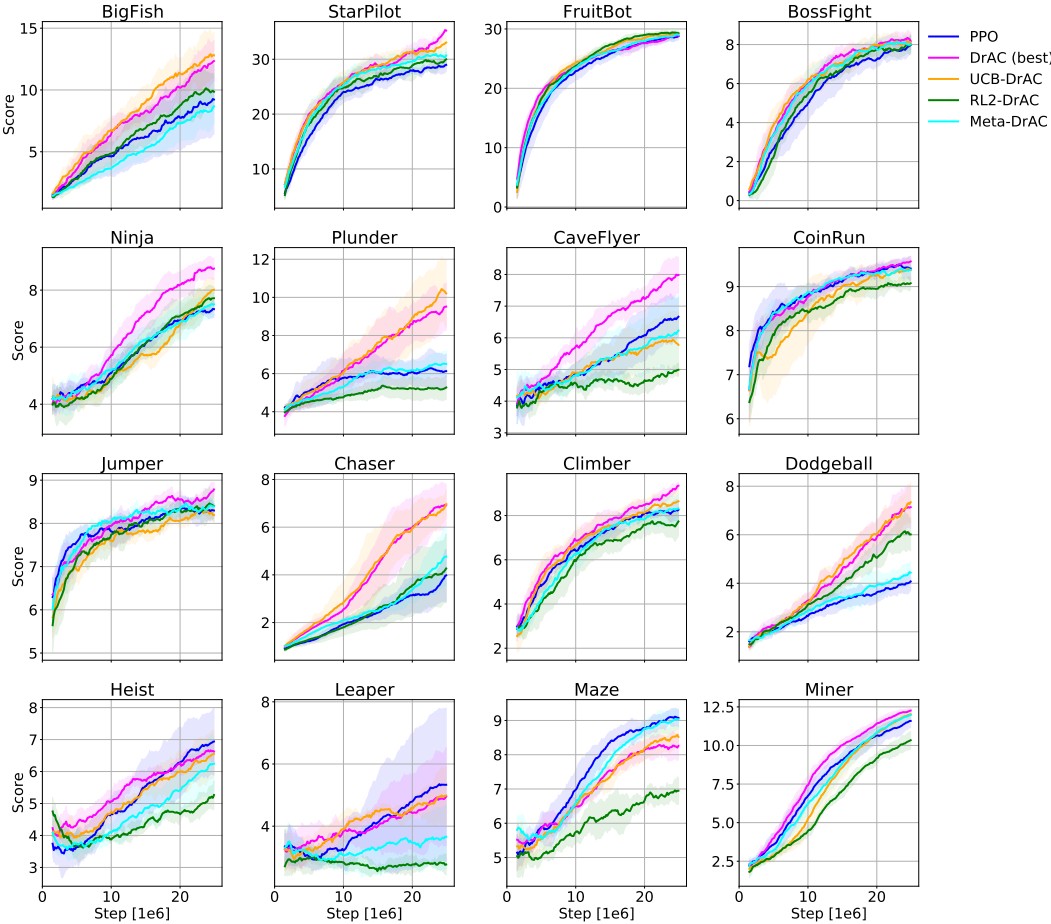

Figure 8: Train performance of various approaches that automatically select an augmentation, namely UCB-DrAC, RL2-DrAC, and Meta-DrAC. The mean and standard deviation are computed using 10 runs.

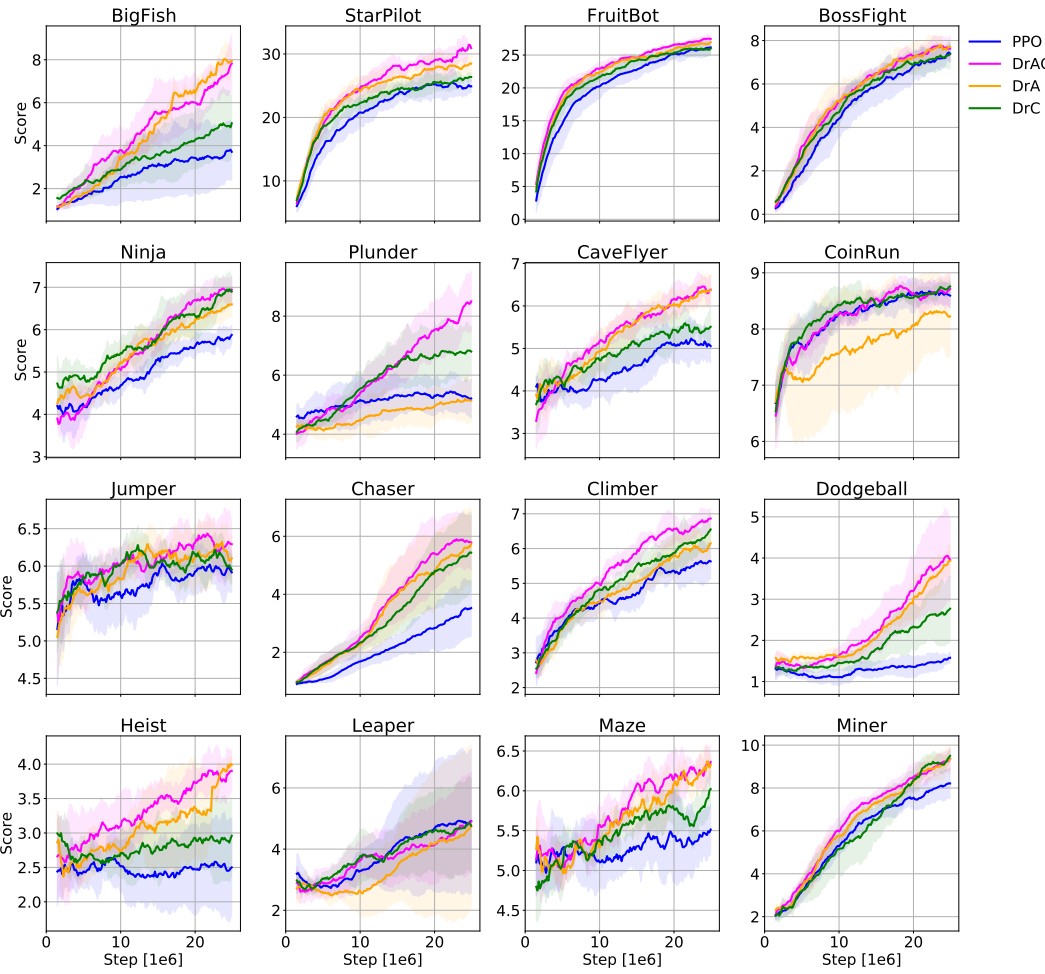

Figure 9: Test performance of PPO, DrAC and its two ablations DrC and DrA (all with the best augmentation for each game) that only regularize the critic or the actor, respectively. The mean and standard deviation are computed using 10 runs.

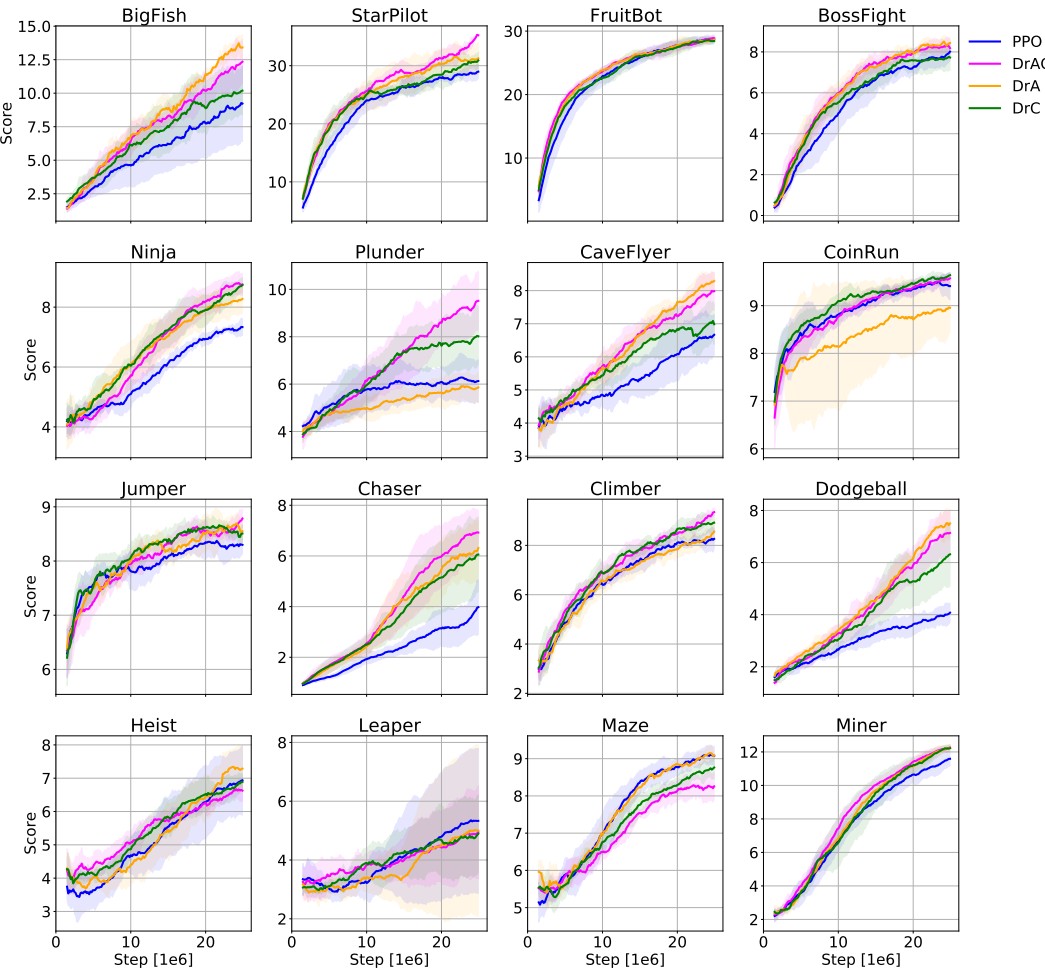

Figure 10: Train performance of PPO, DrAC and its two ablations DrC and DrA (all with the best augmentation for each game) that only regularize the critic or the actor, respectively. The mean and standard deviation are computed using 10 runs.

# L  DeepMind Control Suite Experiments

For our DMC experiments, we ran a grid search over the learning rate in $[1e-4, 3e-4, 7e-4, 1e-3]$, the number of minibatches in $[32, 8, 16, 64]$, the entropy coefficient in $[0.0, 0.01, 0.001, 0.0001]$, and the number of PPO epochs per update in $[3, 5, 10, 20]$. For Walker Walk and Finger Spin we use 2 action repeats and for the others we use 4. We also use 3 stacked frames as observations. For Finger Spin, we found 10 ppo epochs, 0.0 entropy coefficient, 16 minibatches, and 0.0001 learning rate to be the best. For Cartpole Balance, we used the same except for 0.001 learning rate. For Walker Walk, we used 5 ppo epochs, 0.001 entropy coefficient, 32 minibatches, and 0.001 learning rate. For Cheetah Run, we used 3 ppo epochs, 0.0001 entropy coefficient, 64 minibatches, and 0.001 learning rate. We use $\gamma = 0.99$, $\gamma = 0.95$ for the generalized advantage estimates, 2048 steps, 1 process, value loss coefficient 0.5, and linear rate decay over 1 million environment steps. We also found crop to be the best augmentation from our set of eight transformations. Any other hyperaparameters not mentioned here were set to the same values as the ones used for Procgen as described above.

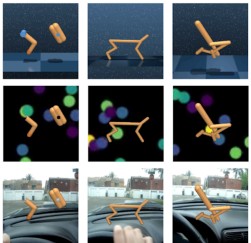

Figure 11: DMC environment examples. Top row: default backgrounds without any distractors. Middle row: simple distractor backgrounds with ideal gas videos. Bottom row: natural distractor backgrounds with Kinetics videos. Tasks from left to right: Finger Spin, Cheetah Run, Walker Walk.

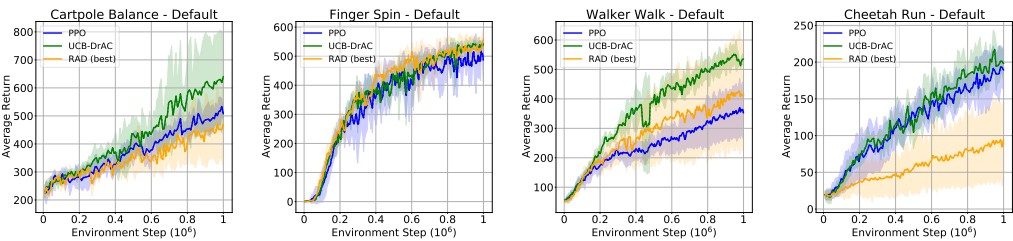

Figure 12: Average return on DMC tasks with default (*i.e.* no distractor) backgrounds with mean and standard deviation computed over 5 seeds. UCB-DrAC outperforms PPO and RAD with the best augmentations.

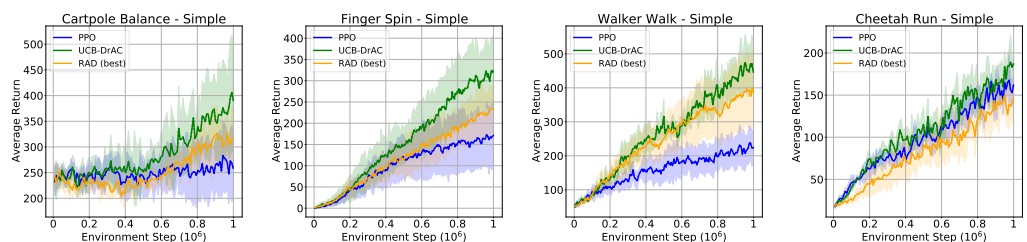

Figure 13: Average return on DMC tasks with simple (*i.e.* synthetic) distractor backgrounds with mean and standard deviation computed over 5 seeds. UCB-DrAC outperforms PPO and RAD with the best augmentations.