# OpenReview forum: "Automatic Data Augmentation for Generalization in Reinforcement Learning"
_NeurIPS.cc/2021/Conference — NeurIPS 2021 Poster_

### Official Review · Reviewer_FoA8 · 2021-07-06

**Rating:** 7
**Confidence:** 3

**Summary:**

The paper proposes a framework for automatic data augmentation in deep RL. It proposes three algorithms based on 1) UCB 2) RL2 3) meta-learning, combining PPO and regularization of both the actor and critic to make sure that the policy and value function are invariant to the transformation corresponding to the augmentation. Experiments on ProcGen and the DMC with distractors show that the UCB-based algorithm (UCB-DrAC) outperforms vanilla deep RL and previous data augmentation baselines and is competitive with the best augmentation type chosen post hoc.

**Limitations And Societal Impact:**

# Limitations #

The authors mention one limitation of the work in the conclusion, that only one augmentation is chosen. I agree that this is a good direction for future work, as in practice many augmentations are often used. Another possible limitation is that the choice of the augmentation is currently based on only the reward signal, but intuitively the state (image) would also be helpful for picking the augmentation.

# Societal Impact #

The authors do not discuss any potential societal impact. I think this is fine since the paper is methodological, and I'd like to mention the following paper on the societal impacts of deep RL.
- Whittlestone, Jess and Arulkumaran, Kai and Crosby, Matthew. The Societal Implications of Deep Reinforcement Learning. JAIR 70, 2021.


**Main Review:**

# Originality #

The paper combines existing ideas to create a novel framework for automatic data augmentation in RL: 1) UCB, RL2, MAML 2) regularizing the value function. Thus, in my opinion, this work has sufficient novelty.

# Quality #

The paper does not contain theory, aside from some intuition for why we should include regularization. I do not consider this a major weakness since the paper is primarily an applied work, but I do think there may be some theoretical support behind regularization, as TRPO does something very similar.

With respect to the empirical results, I found the results to be fairly convincing. The authors use two benchmarks, ProcGen and DCM,  and compare to recent work for generalization in deep RL, both based on data augmentation and other techniques. Ablations are done on the proposed algorithm showing that both the regularization and UCB are necessary for improvements over baselines. The performance of UCB-DrAC is comparable to that of choosing the best augmentation post hoc, with lower sample efficiency.

I would've liked to see more discussion comparing the three proposed algorithms, in particular in relation to the assumption that there is a single best augmentation. If there is a single best augmentation (that happens to be in the set of eight transformations), then it is not surprising that UCB-DrAC works best, as it has the fewest parameters. Do the authors think that that is the case, or could it be that the higher expected sample complexity of RL2-DrAC and Meta-DrAC require training for more timesteps than was done in the paper?

# Clarity #

Overall, the paper is clear, and the results were presented well. However, there are a few places where edits are recommended:
- The description of RL2-DrAC is a little incomplete, it is not clear what the input to the meta-learner is from the main text. After looking at the appendix, I see that it is the sequence of augmentations, not the sequence of states as in the original RL2. I found this to be a bit confusing, so I would suggest the authors to clarify in the final draft.
- The second paragraph of section 4.5 seems a little out of place. Should it have been earlier?

# Significance #

The significance of this work is that it is the first attempt at an automatic data augmentation framework for deep RL. It shows competitive performance to choosing the best augmentation after running all of them, thus leading to improved sample efficiency. As UCB-DrAC seems fairly simple to implement, I think it would be useful for practitioners.

# Updated review #

After reading the responses and reviews, I have decided to keep my score, with the understanding that the authors include more discussion of the assumptions of each algorithm in the final draft.

**Time Spent Reviewing:**

5

---

> ### Author Response · Authors · 2021-08-09
> **Response to Reviewer FoA8**
>
> Thank you for the constructive feedback. We were glad to hear you think our work has “sufficient novelty” and that “it would be useful for practitioners”.
>
> *“Discussion comparing the three proposed algorithms”*
>
> As mentioned in Section 6 and was also noted by one of the reviewers, Meta-DrAC can only express a rather limited class of functions which might be able to generate augmentations similar to random-conv or color-jitter, but likely not crop, which requires a more structured transformation of the input. Overall, we found crop to be the most effective on the Procgen, so we wouldn’t necessarily expect Meta-DrAC to work as well as UCB-DrAC or RL2-DrAC on these environments. We believe it is possible that RL2-DrAC is harder to train than UCB-DrAC in general because it has more parameters, so it has a higher sample complexity.
>
> We also agree that studying when the assumption that there is a single best augmentation holds and how that affects the effectiveness of these algorithms could be a fruitful direction for future work.
>
>
> *“Details of RL2-DrAC”*
>
> Thank you for noting this. We will clarify the details of this algorithm in the final draft.
>
> *“Paragraph in section 4.5”*
>
> Thank you for pointing this out. You are right, that paragraph would be better placed in section 4.1, which we will adjust in the revised draft.
>
>
> *“Base the choice of augmentation on the state in addition to the reward”*
>
> Thank you for this suggestion -- we agree that using the state to select an augmentation could provide additional benefits. However, we believe this is outside the scope of this paper and hope to see this investigated in future work.
>
>
> *“Societal impact”*
>
> Thank you for bringing this to our attention. We will add a discussion of the societal impacts with a reference to the paper you mentioned.

---

> > ### Comment · Reviewer_FoA8 · 2021-08-18
> > **Thank you to the authors for your response**
> >
> > I have decided to keep my original score.

---

### Official Review · Reviewer_jCPN · 2021-07-13

**Rating:** 7
**Confidence:** 4

**Summary:**

Summary: Reinforcement learning has been shown to overfit to training environments. Some recent works (laskin et al, kostrikov et al) have shown data-augmentation to be an effective technique in curbing the overfitting issue. However, different games have different biases and hence require different data augmentation. This paper proposes to use UCB algorithm to automatically select a good data augmentation (from a list of data augmentations) for a game. However, naively applying data augmentation to RL algorithms like PPO isn’t theoretically sound. To address this problem, the paper introduces novel policy and value regularization. Overall, the paper contains extensive experiments and ablation studies.

**Limitations And Societal Impact:**

It's not discussed in the paper.

**Main Review:**

Originality: The paper is a mix of original ideas as well as novel combinations of well known techniques. The paper introduces novel policy and value regularization to combine data augmentation with PPO in a theoretically sound manner. In addition, it uses UCB algorithm to select appropriate data augmentation for a game to be used during the policy training.

Quality: The claims are empirically well supported, the experiments and the analysis are extensive and proper ablations are considered.

Clarity:  The paper is clearly written and well organized.

Significance: The results are important in a way as the paper suggests a theoretically sound and a stable way to use data augmentation with RL.

Minor point: I wouldn’t call the method SOTA as DAAC/iDAAC gets better results (Raileanu et al). But that didn’t affect my final score as I feel this method is complementary to DAAC/iDAAC and it fixes (or atleast makes a good attempt at fixing) the issue of stability wrt data-augmentation in RL.

References:
Michael Laskin, Kimin Lee, Adam Stooke, Lerrel Pinto, Pieter Abbeel, Aravind Srinivas. Reinforcement Learning with Augmented Data. NeurIPS 2020.
Ilya Kostrikov, Denis Yarats, Rob Fergus. Image Augmentation is all you need. ICLR 2021.
Roberta Raileanu, Rob Fergus. Decoupling Value and Policy for Generalization in Reinforcement Learning. ICML 2021.

**Time Spent Reviewing:**

1 hr

---

> ### Author Response · Authors · 2021-08-09
> **Response to Reviewer jCPN**
>
> Thank you for the positive feedback. We were delighted that you found that our paper introduces “a theoretically sound and stable way to use data augmentation in RL”,  and that “the claims are empirically well supported, the experiments and the analysis are extensive and proper ablations are considered”.
>
> *“SOTA on Procgen”*
>
> Indeed, since submitting our paper, other approaches have been proposed which outperform it on Procgen. We will remove the SOTA claims from the paper. We also agree the method is complementary to IDAAC.

---

### Official Review · Reviewer_Toxj · 2021-07-16

**Rating:** 5
**Confidence:** 3

**Summary:**

This paper claims that the current data augmentation methods in RL relies on expert knowledge and large number of transformation methods, and proposes three methods that can automatically find a useful augmentation for a given RL task.
The paper introduce a way of using data augmentation with actor-critic algorithms under settings where the input observation has images.


**Limitations And Societal Impact:**

yes

**Main Review:**

Originality: This paper deals with the data augmentation problem in RL. While the problem setting is not new, the method proposed in this paper do have some merits in automatically learning the augmentation method to reduce the computational requirements. This paper also provides different perspectives on data augmentation, and provided three methods to augment data correspondingly. In the related work, it also discussed adequate paper and differs itself from them.

Quality: This paper technically sound under the scope that the input observations are images.  This paper conducted a bunch of experiments under different environments to validate the effectiveness of their proposed method. However, it seems like in the experiments the input have to be images so that the transformations (candidate data augmentation methods) are applicable. It would be more interesting to see how this method could generate to scenarios where the inputs are vectors (as in many real world cases).  However, I did not find any discussions on this issue in this paper.
Clarity: This paper is clearly written and well organized.

Significance: The results seems to be less important considering the input have to be images.  But I do appreciate that in the experiment part, this paper has investigated the robustness, which is corresponding to their motivations mentioned in the introduction.


**Time Spent Reviewing:**

1

---

> ### Author Response · Authors · 2021-08-09
> **Response to Reviewer Toxj**
>
> Thank you for the valuable feedback, for qualifying our paper as “technically sound”, and for appreciating our robustness analysis.
>
> It seems like the main reason preventing you from wholeheartedly recommending our paper for acceptance is the fact that we only show results on settings where the input observations are images. Below we explain why we made this choice.
>
> First of all, note that the method we propose is orthogonal to the type of input and there is no reason to think it cannot be applied in cases where the observations are represented by vectors (or something else). To adapt our method to other types of inputs, one only needs to use a set of augmentations which are appropriate for that case.
>
> We would also like to point out that, as noted by Calli et al. 2017, Wang et al. 2019, Karpathy et al. 2021 etc., learning from images is well-suited for real-world applications such as self-driving cars or robotic manipulation tasks since images are typically cheaper to collect than other sensory data.
>
> Moreover, data augmentation can be particularly important when training agents directly from images (rather than low dimensional states) since this setting requires more samples to learn good representations (as recently shown by Srinivas et al. 2020, Kostrikov et al. 2020, and Laskin et al. 2020).
>
> Finally, we aren’t aware of any well-established generalization benchmarks where the observations are represented by vectors, so we chose to evaluate our approach on Procgen and DMC with distractors which have been widely used to probe generalization in RL.
>
> We hope our response is sufficient for you to reconsider your assessment of the paper. But please don’t hesitate to let us know if you have any outstanding concerns so that we have an opportunity to address them.
>
> References:
> [1]  Calli, B., Singh, A., Bruce, J., Walsman, A., Konolige, K., Srinivasa, S., ... & Dollar, A. M. (2017). Yale-CMU-Berkeley dataset for robotic manipulation research. The International Journal of Robotics Research, 36(3), 261-268.
> [2] Wang, A., Kurutach, T., Liu, K., Abbeel, P., & Tamar, A. (2019). Learning robotic manipulation through visual planning and acting. arXiv preprint arXiv:1905.04411.
> [3] Karpathy, A., Workshop on Autonomous Driving Keynote Talk, CVPR, 2021.
> [4] Srinivas, A. (2020). Reinforcement learning with augmented data. arXiv preprint arXiv:2004.14990.
> [5] Kostrikov, I., Yarats, D., & Fergus, R. (2020). Image augmentation is all you need: Regularizing deep reinforcement learning from pixels. arXiv preprint arXiv:2004.13649.
> [6] Laskin, M., Lee, K., Stooke, A., Pinto, L., Abbeel, P., & Srinivas, A. (2020). Reinforcement learning with augmented data. arXiv preprint arXiv:2004.14990.

---

### Official Review · Reviewer_CGaL · 2021-07-16

**Rating:** 7
**Confidence:** 4

**Summary:**

This paper presents a method for improving generalization in reinforcement learning using data augmentation. It identifies a key theoretical limitation of previous work in the naive application of data augmentation to actor critic methods and proposes a solution that avoids biased estimation of the policy gradient objective while still allowing for regularization of the policy and value function. The resulting loss is paired with three distinct approaches for identifying an optimal data augmentation strategy. The best of these approaches, a bandit algorithm, obtains SOTA performance (relative to published works) on the Procgen environment.


**Limitations And Societal Impact:**

While I don't believe this work is likely to lead to any direct negative societal impacts, the authors should include a statement saying as much in the text.

**Main Review:**

Strengths:

- The paper provides a well-motivated and justified means of applying data augmentation to improve generalization in reinforcement learning.

- The use of data augmentation to promote invariance in the value function and policy directly, rather than indirectly via applying the data augmentation to the inputs used to compute the policy gradient, is a nice way of avoiding the issue of biased gradients in policy gradient learners.

- The UCB method to select data augmentation transformations is a nice addition to promote more effective augmentation policies.

- The paper is well-written and clear, with the experimental settings easy to follow and the theoretical motivation compelling.

- The empirical analysis of different approaches to automatically learning data augmentation is reasonably complete and provides some insight into the robustness of the proposed regularization strategy to possibly-suboptimal augmentation sets.

Weaknesses:

- The proposed method isn’t particularly novel as data augmentation approaches have been widely used in both supervised and unsupervised learning to regularize the representation. Applying data augmentation only to enforce invariance in a representation, as opposed to using it to increase the size of the training dataset, has been deployed previously in papers such as BYOL.

- The meta-learning baseline is unclear: does the meta-learner only have the capacity to apply a single augmentation to the inputs? If so, this seems like a fairly severe limitation. A more fair comparison might have been against a learned Spatial Transformer Network, which has the capacity to apply more complex sets of augmentations.

- The empirical performance indeed outperforms the baselines, but in many instances the performance improvements are relatively modest.
UCB assumes a fixed reward distribution, but as mentioned by the authors the policy improvements of the reinforcement learner mean that the rewards are likely to be highly nonstationary. Additionally, the use of the “training” environment to select the data augmentation, rather than a “validation” environment, seems like it might encourage overfitting.

Questions and Points of Improvement

- There are additional approaches to RL that seek to throw away irrelevant information to improve generalization, such as DBC and PBL. I would be interested in seeing how the method compares to these other approaches that use methods other than data augmentation to improve robustness and generalization.

- The theoretical setup should clarify whether the goal is truly to maximize the expected return over all environments or to maximize some sort of normalized return. For example, if one environment provides significantly more rewards than another, it might make sense to focus policy learning on this environment. This presumably isn’t an issue for the environments under consideration in the empirical analysis because of the shared structure, but may be worth noting for correctness.

- I would have liked to see more justification for the use of training-set performance improvement in selecting a transformation rather than a validation-environment performance improvement as the reward signal for UCB, as it seems the principal goal of this method is not to increase the performance on the training environments, but to improve test-set performance.

- The best distribution of augmentations to maximize generalization performance may not be to greedily select the augmentation that improves the policy improvement step the most. I would be interested in seeing if a similar approach that can identify stochastic optimal policies over augmentations might yield better performance.


**Time Spent Reviewing:**

5

---

> ### Author Response · Authors · 2021-08-09
> **Response to Reviewer CGaL**
>
> Thank you for the thoughtful feedback and for finding our paper “well-written and clear”, “the theoretical motivation compelling”, and our approach “well-motivated and justified”. We address your concerns below.
>
>
> ### Main Concerns:
>
>
> *“The proposed method isn’t particularly novel as data augmentation approaches have been widely used in both supervised and unsupervised learning to regularize the representation”.*
>
> We respectfully disagree with this characterization of our method. First of all, note that we do not directly regularize the representation, but rather the policy and value function, which are specific to the RL setting. Second, in addition to the regularization terms, we also introduce a method for selecting the most suitable augmentation from a given set. As far as we know, this is the first time automatic data augmentation was successfully applied in RL. As we explain in the paper, the RL setting comes with additional challenges (relative to the supervised and unsupervised learning) which need to be addressed in order to use data augmentation in a principled and effective way.
>
>
> *“The meta-learning baseline is unclear: does the meta-learner only have the capacity to apply a single augmentation to the inputs?”*
>
> The meta-learning baseline uses a convolutional network as the augmentation, whose parameters are meta-learned. We discuss that this can only represent a limited class of functions in Section 5, and we leave more complex parameterizations for future work.
>
>
> *“UCB assumes a fixed reward distribution, but as mentioned by the authors the policy improvements of the reinforcement learner mean that the rewards are likely to be highly nonstationary.”*
>
> Note that we use a variant of UCB which alleviates nonstationarity. More specifically, we use a sliding window (containing the most recent data) to compute the statistics used in UCB.
>
>
> *“Additionally, the use of the “training” environment to select the data augmentation, rather than a “validation” environment, seems like it might encourage overfitting.”*
>
> While we agree this would be an important hypothesis to test in future work, we chose to use the same set of environments for training UCB as for all the other methods in order to make the comparison as fair as possible.
>
>
> ### Other Questions:
>
>
> *“Comparisons with DBC and PBL”*
>
> While we agree there are interesting connections between DrAC, DBC, and PBL, and a thorough comparison of these methods would be valuable, we don't think DBC and PBL are necessarily well suited for improving generalization on the Procgen benchmark, which was the main testbed we used to evaluate our method. First of all, neither DBC nor PBL have been previously evaluated on Procgen.
>
> In particular, DBC makes the assumption that the underlying transition function is the same across all environments and also relies on dense rewards to learn good state representations. However, because Procgen games are procedurally generated and many of them have partial observability, the next state (in two different environments) can be different even for the same underlying current state and action, thus breaking the assumption made by DBC. In addition, most tasks in Procgen have sparse rewards, which makes it challenging to learn good representations using DBC.
>
> Similarly, PBL was only evaluated on the training environments, so it wasn't specifically designed to improve generalization to new task instances. Moreover, PBL relies on predicting future latent states in the environment. However, there is no reason to think it can accurately predict future latent states in new environments with partial observability (which is the case when evaluating on Procgen).
>
>
> *“Theoretical setup”*
>
> Thank you for the suggestion. We will discuss this in the revised draft.
>
>
> *“Training v. validation for selecting the transformation”*
>
> As mentioned above, we chose to use the performance on the training environments for selecting the augmentation in order to allow UCB-DrAC to make use of all the available data just like the other approaches (DrAC, RAD etc.). But we think this question should be investigated in future work.
>
>
> *“Stochastic optimal policies over the set of augmentations”*
>
> Absolutely, we also believe there might be cases in which a stochastic policy would be more effective than a deterministic one. As discussed in Section 5, we think this would be a promising direction for future work.
>
>
> We hope our clarifications have adequately addressed your concerns and give you the comfort to fully support our paper with your score.

---

> > ### Comment · Reviewer_CGaL · 2021-08-28
> > **Most issues resolved, updating score**
> >
> > Thanks to the authors for the helpful response. The discussion in section 3.1 highlights that applying data augmentation approaches from supervised learning directly to policy-gradient algorithms is not straightforward and so the existence of both existing data augmentation approaches for RL and of learned augmentation strategies for supervised learning does not mean that the proposed approach for learning augmentation strategies for policy-gradient RL is a straightforward modification to existing methods. I am therefore reasonably convinced by the author's arguments for the novelty of the method. I am also convinced that a comparison to DBC and PBL would not necessarily provide significant information gain given the settings used for the empirical evaluations.
> >
> > I still think that the paper would benefit from a) applying the method to stochastic data augmentation policies, and b) using a train-validation-test split for all methods to select the best augmentations. However, these two points do not prevent me from recommending the paper for acceptance in approximately its current form.

---

> > > ### Author Response · Authors · 2021-08-29
> > > **Thank you**
> > >
> > > We thank the reviewer for engaging in the discussion and for their thoughtful comments. We will do our best to take this feedback into account.

---

### Decision · Program_Chairs · 2021-09-27

**Decision:**

Accept (Poster)

**Comment:**

The paper presents a solid contribution to improve generalization in reinforcement learning through data augmentation. Reviewers appreciated the theoretical motivations provided for the method and the addition of UCB to select transformations. I encourage the authors to make revisions promised in their response and address the issues raised by the reviewers.